# Longitudinal development of category representations in ventral temporal cortex predicts word and face recognition

Marisa Nordt [1,2,3] ✉, Jesse Gomez [4], Vaidehi S. Natu [1], Alex A. Rezai[1], Dawn Finzi [1,5], Holly Kular[1] & Kalanit Grill-Spector [1,6,7]

Regions in ventral temporal cortex that are involved in visual recognition of categories like words and faces undergo differential development during childhood. However, categories are also represented in distributed responses across high-level visual cortex. How distributed category representations develop and if this development relates to behavioral changes in recognition remains largely unknown. Here, we used functional magnetic resonance imaging to longitudinally measure the development of distributed responses across ventral temporal cortex to 10 categories in school-age children over several years. Our results reveal both strengthening and weakening of category representations with age, which was mainly driven by changes across category-selective voxels. Representations became particularly more distinct for words in the left hemisphere and for faces bilaterally. Critically, distinctiveness for words and faces across category-selective voxels in left and right lateral ventral temporal cortex, respectively, predicted individual children's word and face recognition performance. These results suggest that the development of distributed representations in ventral temporal cortex has behavioral ramifications and advance our understanding of prolonged cortical development during childhood.

Recognizing faces and written words is important for our everyday life. While extensive experience with faces starts from birth[1], extensive experience with written words typically begins around the age of 5–6 with the onset of formal reading education. Nonetheless, both skills continue to improve throughout childhood and adolescence[2,3]. Visual recognition is thought to involve the end stage of the human ventral visual processing stream – that is, ventral temporal cortex (VTC). Indeed, VTC contains both (i) clustered regions that are selective to and causally involved in the perception of ecologically relevant categories[4–6] such as faces[7], limbs[8,9], places[10], and words[11], as well as (ii) distributed representations that are reproducibly unique to each category[12–14]. That is, stimuli from different categories generate distinct patterns of response across VTC, even stimuli that are not associated with clustered regions and even when excluding category-selective regions altogether. However, it is unknown if and how distributed category representations in VTC develop longitudinally during childhood and if these developments are linked with behavioral improvements in visual recognition.

Understanding the longitudinal development of distributed representations during childhood is important for three main reasons. First, to date, examinations of the development of distributed VTC responses have been largely confined to cross-sectional studies which

[1]Department of Psychology, Stanford University, Stanford, CA, USA. [2]Department of Child and Adolescent Psychiatry, Psychosomatics and Psychotherapy, Medical Faculty, RWTH Aachen, Aachen, Germany. [3]JARA-Brain Institute II, Molecular Neuroscience and Neuroimaging, RWTH Aachen & Research Centre Juelich, Juelich, Germany. [4]Princeton Neuroscience Institute, Princeton University, Princeton, NJ, USA. [5]Department of Computer Science, Stanford University, Stanford, CA, USA. [6]Neurosciences Program, Stanford University, Stanford, CA, USA. [7]Wu Tsai Neurosciences Institute, Stanford University, Stanford, CA, USA. ✉e-mail: mnordt@ukaachen.de

compared representations in children to adults using a limited number of categories with inconsistent results[15–18]. While some studies have reported that distributed responses to faces, objects, and places in children are similar to adults[15,17], other studies have reported that own-age face representations[17] and word representations[18] are enhanced from childhood to adulthood. Thus, to understand how distributed VTC representations change during childhood it is necessary to measure the longitudinal development of distributed representations within the same children and with many categories. Second, much of the prior developmental research has examined the development of category selectivity within clustered regions selective to ecologically-relevant categories[16,19–25]. However, the development of distributed responses may be different than the development of category representations in clustered regions as different signals contribute to distributed compared to clustered responses. That is, the distributed pattern of response is determined by all values of responses (both high and low) across VTC, whereas in clustered regions the selectivity is driven by the highest responses[26]. Third, examining the relation between the development of distributed VTC responses and the development of face recognition and reading abilities will address a key debate in the field. Researchers argue whether the entire distributed pattern of response across VTC[13] or specific distributed responses over the selective voxels[18,27–29] gives rise to developmental improvements in visual recognition behavior.

How may distributed category representations in VTC develop during childhood?

Prior research has found that clustered regions selective to faces and words become larger and more selective to their respective categories from childhood (4–7 years old) to adolescence (13–17 years old) to adulthood (>18)[16,19–25]. Recent longitudinal research in children has surprisingly found that these increases in face- and word-selectivity in lateral VTC are coupled with decreases in limb-selectivity and, in fact, regions that are limb-selective earlier in childhood become selective to faces and words by adolescence[23]. These findings suggest that changes in category selectivity may affect the nature of distributed VTC representations. One possibility is that developmental increases in category-selectivity would lead to increases in the distinctiveness of distributed responses for categories associated with a clustered region in VTC. For instance, developmental increases in word- and face-selectivity would increase the distinctiveness of distributed responses to words and faces, making distributed responses both more consistent across items of their respective category and more different from distributed responses to items of other categories. This hypothesis also predicts that developmental decreases in limb-selectivity would lead to decreases in the distinctiveness of distributed responses to limbs. A second possibility is that distributed responses in VTC develop for many categories beyond faces, limbs, and words, and also across voxels that don't have a strong preference for any category. This hypothesis predicts that developmental changes will be observed also for categories, such as numbers, for which there is no clustered region in VTC, and also across voxels that have no specific selectivity to any category. A third possibility is that because clustered regions selective to faces, words, and limbs only constitute a minority of VTC voxels[18], changes in these regions will have little effect on large-scale distributed patterns of response. Thus, this hypothesis predicts no significant development of distributed VTC responses during childhood, consistent with cross-sectional data reporting adult-like distributed responses to faces, objects, and scenes by age 7[15,17].

These hypotheses also make different behavioral predictions. The first hypothesis predicts that if distributed representations of faces and words become more distinct over childhood development, then face recognition and reading performance will concurrently improve. This hypothesis also predicts that behavioral changes in face recognition and reading will be coupled with specific increases in the distinctiveness of distributed responses over the category-selective voxels. The second hypothesis predicts that developmental increases in the distinctiveness of distributed responses across the entire VTC rather than over category-selective voxels will be coupled with behavioral improvements in face recognition and reading, and this relationship may hold even when voxels of category-selective regions are excluded from the distributed response across VTC. In contrast, the third hypothesis does not predict a relationship between the development of face recognition and reading performance and their respective distributed representations in VTC as the latter are predicted to stay stable during childhood development.

Here, we test these predictions using longitudinal measurements of distributed responses to many visual categories as well as behavioral assessments of face recognition and reading in the same school-age children over several years. These longitudinal measurements are crucial not only for tracking the development of brain and behavior within the same child over several years but also for evaluating the rate of development of distributed responses both for categories with and without clustered regions of strong selectivity in VTC. Thus, we collected longitudinal functional magnetic resonance imaging (fMRI) and behavioral data in 29 school-age children over a span of 1 to 5 years (mean ± SD: 3.75 ± 1.5 years, 4.4 ± 1.92 sessions per child) totaling 128 fMRI sessions and 146 behavioral datasets (Fig. S1A). During the fMRI experiment, children viewed 1440 images from 10 categories spanning 5 domains of ecological relevance (Fig. S1B). These include characters (pseudowords, numbers), faces (adult faces, child faces), body parts (headless bodies, limbs), objects (cars, string instruments), and places (houses, corridors). We measure in each session distributed responses for each of the 10 categories across VTC and examine if they change as children aged. To relate brain development to visual recognition performance, we measure in the same children their face recognition and reading performance outside the scanner (face recognition: 29/29 children, 2.83 ± 1.0 sessions per child collected over 3.38 ± 1.5 years, reading: 26/29 children, 2.21 ± 1.1 sessions per child collected over 2.86 ± 1.19 years). Then, we test if there is a relationship between behavioral and brain development. We find that distributed category representations, especially across category-selective voxels of VTC both strengthen and weaken across childhood development. Notably, distinctiveness for words and faces across category-selective voxels in left and right lateral VTC, respectively, predicts children's word and face recognition performance.

## Results

### How do category representations in VTC develop longitudinally?

To assess the nature of distributed category representations in children, we computed the distributed pattern of responses for each category. As face-, limb-, and word-selective regions are located in the lateral aspect of VTC[7,8,11,30], we divided VTC into its lateral and medial partitions and because the development of word- and face-selectivity varies across hemispheres[16,18,21,31], we measured in each child and session distributed responses to each of the 10 categories, separately for lateral and medial VTC in each hemisphere. Vectors of distributed responses to each category – also called multivoxel patterns (MVPs) – were computed independently for each of the two functional runs in each session in which participants viewed different images. We then calculated correlations between all pairs of MVPs (run-1 to run-2), resulting in a 10×10 representational similarity matrix (RSM[14], for each child and session (Fig. 1a). On-diagonal values in the RSM quantify how similar distributed responses are across different images of the same category, and off-diagonals quantify how similar distributed responses are to images of different categories. Examining individual RSMs revealed that even in young children there is category structure in distributed VTC responses as on-diagonal values are positive and higher than off-diagonal values (Fig. 1a).

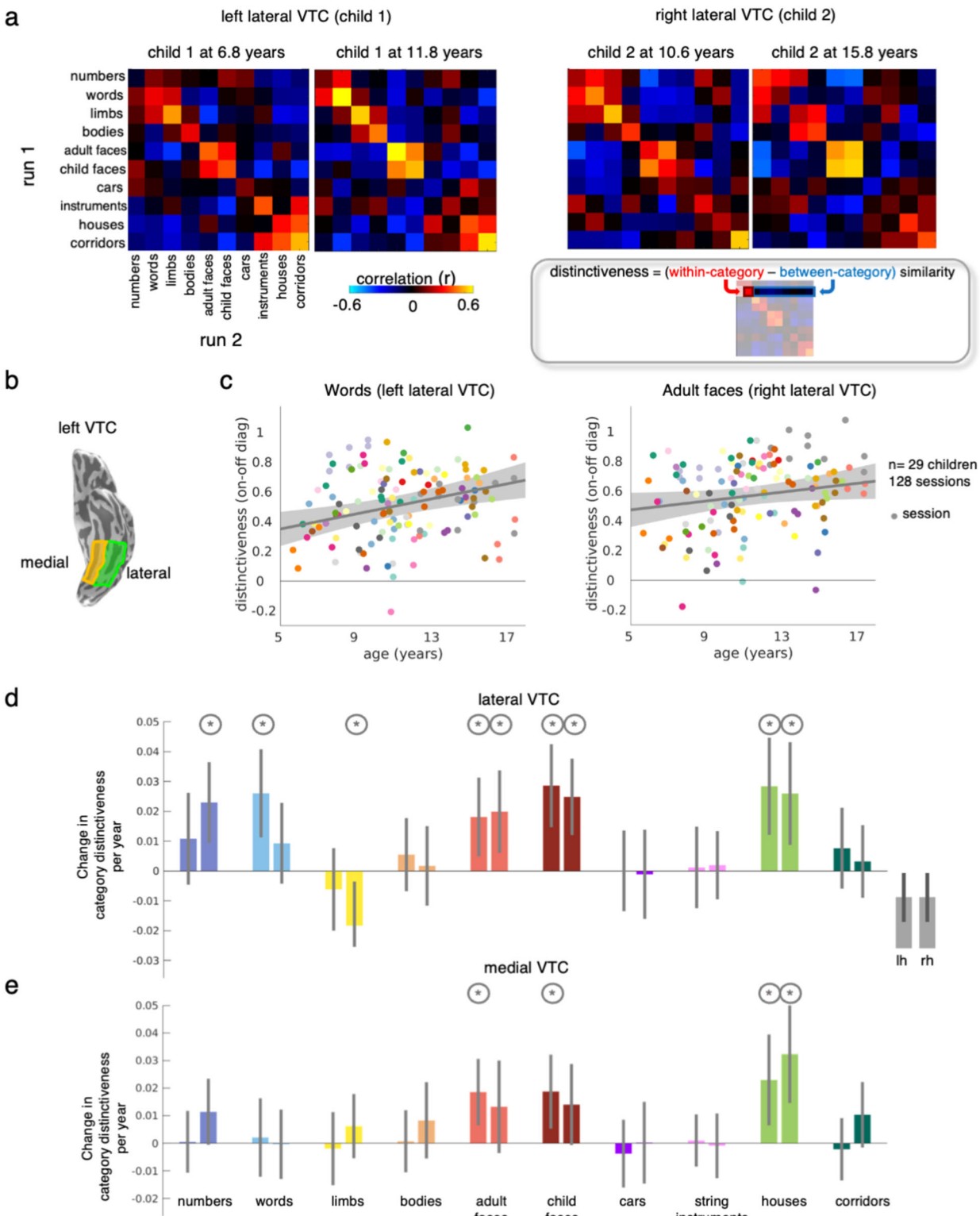

**Fig. 1 | Differential longitudinal development of category representations in children's ventral temporal cortex (VTC). a** Representational similarity matrices (RSM) of left and right lateral VTC in individual sessions of two children at different ages. Gray box: Schematic illustrating how distinctiveness is computed for each category. **b** Left lateral (green) and medial (yellow) VTC on the ventral inflated surface of an example participant. **c** Scatter plots illustrate the relationship between distinctiveness and age. Gray line: Linear mixed model (LMM) prediction of distinctiveness by age (random intercept model with participant as a random effect). Shaded gray: 95% confidence interval (CI) of the slope. Participants are coded by color. Each dot is a session. Note that the statistics in the text and the data in (**d**)

report developmental results from a LMM that includes two factors: age and tSNR. **d** LMM slopes indicating a change in distinctiveness per year in lateral VTC (LMM relating distinctiveness to age, with tSNR as an independent factor, and participant as random effect, *n* = 128 sessions, 29 children) for each category; for space, we refer to pseudowords as words. Error bars: 95% confidence interval (CI) of the slope. If the CI does not cross the y = 0 line, the change in distinctiveness is significantly different than 0. Asterisks: significant development (*p* < 0.05). Circles around asterisks: significant development after FDR-correction to adjust for multiple comparisons. **e** same as D but for medial VTC. Full statistics are reported in Tables S1–2.

To quantify category information, we used the RSM to measure the distinctiveness of distributed responses for each category. The distinctiveness of a category is defined as the difference between within-category similarity and between-category similarity (Fig. 1a, gray box); distinctiveness ranges from −2 to 2. Higher distinctiveness indicates that distributed responses to a category are highly similar across different images of the category and highly dissimilar from distributed responses to other categories. We measured distinctiveness separately for each category and hemisphere in each of the 128 sessions. Then, we tested if category distinctiveness develops over childhood using linear mixed models (LMMs) relating category distinctiveness and age, with the participant as a random factor (Fig. 1c).

To ensure that developmental effects are not driven by differences in scan quality across age, we first tested whether motion during scanning and time-series signal-to-noise ratio (tSNR) contribute to measures of distinctiveness. Checking for effects of motion was included after our data had already been corrected for motion (see Methods) to further ensure that our results are not impacted by motion-related artifacts that are missed by motion correction algorithms. Adding motion as a predictor to the LMM did not significantly contribute to the model fit except for distinctiveness for string instruments (no evidence for a significant contribution of age to the distinctiveness of string instruments with or without adding motion). Adding tSNR as a predictor to the LMM contributed to the model independent from age for several categories. Thus, we include tSNR as an additional predictor in the LMM (Tables S1–2) and we report age-related changes in distinctiveness that are independent from tSNR.

Example scatter plots show the distinctiveness for pseudowords (Fig. 1c-left) and adult faces in lateral VTC (Fig. 1c-right) as a function of a child's age. As you can see in these examples, distinctiveness for pseudowords in the left lateral VTC and distinctiveness for adult faces in the right lateral VTC steadily increases from age 5 to 17. The slope of the LMM summarizes the development of category distinctiveness with age. A positive slope indicates that category distinctiveness increases from age 5 to 17 and a negative slope indicates that category distinctiveness decreases across childhood.

Results reveal differential development of category distinctiveness in VTC that varied by category and hemisphere (Fig. 1d, e) and that was not limited to categories with a clustered region. We first examined development in lateral VTC. Consistent with the first hypothesis, we find increases in distinctiveness for pseudowords and faces and decreases in distinctiveness for limbs: Distinctiveness for pseudowords increased significantly with age in the left lateral VTC ($\beta_{age}$ = 0.026, t(125) = 3.49, $p_{FDR}$ = 0.0038, Fig. 1c, d), but there was no evidence for significant development in right lateral VTC ($\beta_{age}$ = 0.009, t(125) = 1.36, $p_{FDR}$ = 0.32). Distinctiveness for both adult faces (left: $\beta_{age}$ = 0.018, t(125) = 2.72, $p_{FDR}$ = 0.019; right: $\beta_{age}$ = 0.0199, t(125) = 2.85, $p_{FDR}$ = 0.015) and child faces (left: $\beta_{age}$ = 0.029, t(125) = 4.06, $p_{FDR}$ = 0.0017; right: $\beta_{age}$ = 0.025, t(125) = 3.84, $p_{FDR}$ = 0.002) increased significantly with age in both hemispheres (Fig. 1c, d). In contrast to these developmental increases, distinctiveness for limbs decreased significantly with age in the right hemisphere ($\beta_{age}$ = −0.018, t(125) = −2.45, $p_{FDR}$ = 0.035) while distinctiveness for bodies remained largely unchanged from age 5 to 17. Consistent with the second hypothesis, we find increases in distinctiveness even for categories that do not have a clustered region in lateral VTC. Specifically, (i) distinctiveness for numbers increased significantly in the right lateral VTC ($\beta_{age}$ = 0.023, t(125) = 3.36, $p_{FDR}$ = 0.004); while there was no evidence for significant development in left lateral VTC ($\beta_{age}$ = 0.011, t(125) = 1.39, $p_{FDR}$ = 0.32) and (ii) distinctiveness for houses increased significantly bilaterally (left: $\beta_{age}$ = 0.028, t(125) = 3.45, $p_{FDR}$ = 0.0038, right: $\beta_{age}$ = 0.026, t(125) = 2.98, $p_{FDR}$ = 0.011). We find no evidence for other significant age-related changes (Table S1). We next examined development of distinctiveness

in medial VTC. Consistent with the first hypothesis, we find increases in distinctiveness for houses in left and right medial VTC (left: $\beta_{age}$ = 0.023, t(125) = 2.76, $p_{FDR}$ = 0.034, right: $\beta_{age}$ = 0.032, t(125) = 3.61, $p_{FDR}$ = 0.009). Consistent with the second hypothesis, we also find development for a category without a clustered region in medial VTC: Distinctiveness for adult and child faces increased significantly in the left hemisphere in medial VTC (adult faces: $\beta_{age}$ = 0.0186, t(125) = 3.048, $p_{FDR}$ = 0.028, child faces: $\beta_{age}$ = 0.0187, t(125) = 2.76, $p_{FDR}$ = 0.034). We find no evidence for other significant development in medial VTC (Table S2). Overall, these analyses reveal differential development of distributed category representation in VTC from age 5 to 17.

## Which voxels drive the development of distributed representations?

We next asked: which voxels in VTC drive the development of category representation? Is the development of distinctiveness driven by category-selective voxels, non-selective voxels, or both? We reasoned that if changes in distributed responses are driven by the development of category selectivity, then the development of category distinctiveness will be evident in the selective but not in the non-selective voxels of lateral VTC. Alternatively, voxels that are selective may be already developed, predicting that the non-selective voxels are driving the observed development of category distinctiveness. A third possibility is that category information is carried by the relative response across the entire neural population and in fact, there is nothing special about the selective voxels[13]. This hypothesis predicts that the development of category distinctiveness is driven by all voxels of VTC, including both the selective and non-selective voxels.

To test these predictions, we separated lateral and medial VTC into two sets of voxels each – those which were category-selective, and the rest that were not selective to any category. That is, for each session, we first identified lateral (or medial) VTC voxels that were selective (t > 3, voxel-level, Methods) for each of the 10 categories, and then took the union of these voxels across the 10 categories to generate the set of all category-selective voxels. Non-selective voxels were defined as the remainder of lateral (or medial) VTC voxels and were not selective to any of these categories. We next computed LMMs relating distinctiveness to age, with tSNR as an independent factor, in these two subsets of voxels (see Tables S3–6 for all model parameters). We report the slope of the LMM indicating the effect of age for each category in Fig. 2. We first examine the results in lateral VTC. Although there are overall fewer selective voxels (left: 39.24% ± 11; right: 34% ± 11%) than non-selective voxels (left: 60.76% ± 11; right: 66% ± 11%) we find a significant development of category distinctiveness across the union of selective voxels of lateral VTC (Fig. 2a-maroon & pink bars), but no evidence for significant development of distinctiveness across the non-selective voxels of lateral VTC (Fig. 2a-gray bars). These findings were not due to higher tSNR in the union of selective voxels compared to the non-selective voxels (no evidence for a significant effect of voxel subset in left lateral VTC: $\beta_{subset\_Selective}$ = 1.8, t(254) = 1.34, p = 0.18 and higher tSNR in non-selective vs. selective voxels in right lateral VTC: $\beta_{subset\_Selective}$ = −3.4, t(254) = −2.47, p = 0.01, LMM with binary predictor of voxel subset). Additionally, we repeated the analyses in subsets of the union of the selective and the non-selective voxels that were matched for the variance explained and for the number of included voxels. This analysis showed the same pattern of significant development of distinctiveness for the union of the selective voxels, and no evidence of significant development for the non-selective voxels, except for a significant increase in distinctiveness for numbers (Fig. S2) suggesting that the lack of development in the non-selective voxels is not driven by poorly responding voxels.

In fact, the development of category distinctiveness in the subset of selective voxels largely replicates the findings when considering all

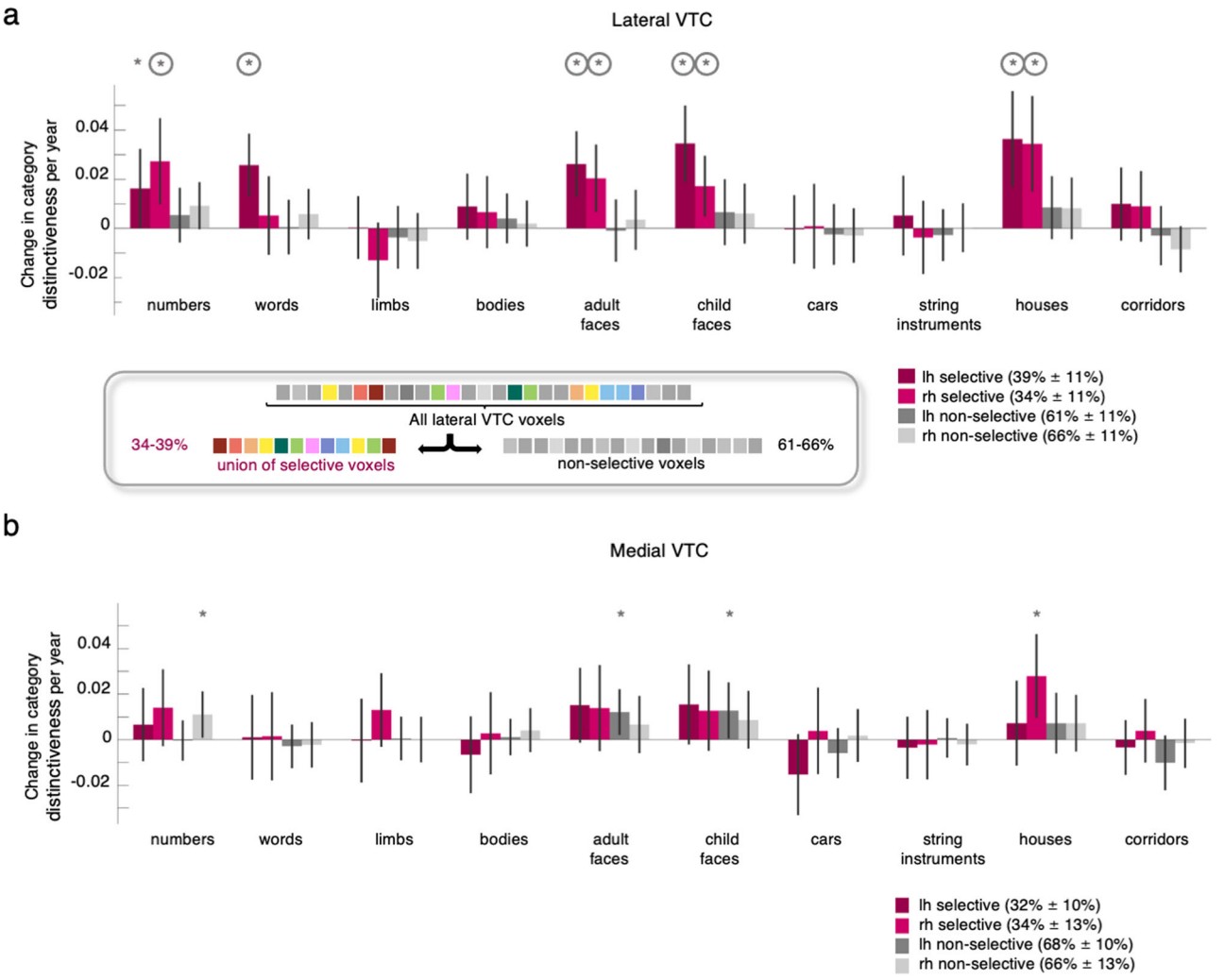

**Fig. 2 | Development of distributed representation in the selective voxels in lateral VTC. a** Bars indicate the change in category distinctiveness per year (LMM relating distinctiveness to age and tSNR, with participant as a random effect, $n = 128$ sessions, 29 children) in different subsets of voxels of lateral VTC. *Maroon bars:* the union of voxels that were selective to one of the 10 categories in lateral VTC. Category-selectivity was computed by contrasting responses to a category vs. all other categories except the other category from the same domain (e.g., numbers vs. all other categories except words). A voxel was defined as selective to a category when t > 3. Overall ~36.5% of lateral VTC voxels were selective to one of the categories (see schematic in box). *Gray bars:* the remainder, non-selective voxels of lateral VTC that were not selective to any of these categories. *Darker colors:* left hemisphere. *Lighter colors:* right hemisphere. *Error bars:* 95% CI. If the CI does not cross the y = 0 line, the change in distinctiveness is significantly different than 0. Asterisks indicate significant development (*p* < 0.05). Circles around asterisks indicate significant development after FDR-correction to adjust for multiple comparisons. **b** Same as A but for medial VTC. Full statistics are reported in Tables S3–6.

lateral VTC voxels (compare Figs. 1d and 2a). That is, in the union of selective voxels in the left hemisphere, we find a significant increase in the distinctiveness for pseudowords ($\beta_{age} = 0.026$, t(125) = 3.99, $p_{FDR} = 0.002$), faces (adult: $\beta_{age} = 0.026$, t(125) = 3.88, $p_{FDR} = 0.002$; child: $\beta_{age} = 0.034$, t(125) = 4.39, $p_{FDR} = 0.0009$), houses ($\beta_{age} = 0.036$, t(125) = 3.69, $p_{FDR} = 0.003$) and a small increase in distinctiveness for numbers ($\beta_{age} = 0.016$, t(125) = 1.998, $p_{FDR} = 0.21$), which was not significant after FDR-correction. In the right hemisphere, we find a significant increase in the distinctiveness for numbers ($\beta_{age} = 0.027$, t(125) = 3.08, $p_{FDR} = 0.017$), faces (adult: $\beta_{age} = 0.02$, t(125) = 2.95, $p_{FDR} = 0.02$; child: $\beta_{age} = 0.017$, t(125) = 2.75, $p_{FDR} = 0.03$), and houses ($\beta_{age} = 0.034$, t(125) = 3.49, $p_{FDR} = 0.005$). We find no evidence for a significant decrease in the distinctiveness for limbs in this subset of voxels ($\beta_{age} = -0.013$, t(125) = -1.67, $p_{FDR} = 0.33$). In contrast to these effects in the union of the selective voxels, we find no evidence for significant changes in distinctiveness for any category in the non-selective voxels of lateral VTC (Fig, 2A-gray bars, Table S4). We tested if these results change if the union of the selective voxels and the non-

selective voxels are defined using different thresholds (Methods). We found that the results are largely robust across a range of t-thresholds defining selectivity (Fig. S3).

We next examined the development of distinctiveness in these subsets of voxels in medial VTC. As in lateral VTC, the union of the selective voxels comprised less voxels compared to the non-selective voxels (see legend Fig. 2b). Examining changes in distinctiveness in the union of the selective voxels revealed an increase in distinctiveness for houses in right medial VTC ($\beta_{age} = 0.028$, t(125) = 3.03, $p_{FDR} = 0.12$), which was not significant after FDR-correction (Fig. 2b-maroon & pink bars). In the non-selective voxels (Fig. 2b-gray bars), there were increases in distinctiveness for faces in the left and for numbers in the right hemisphere, which were not significant after FDR-correction (adult faces: $\beta_{age} = 0.012$, t(125) = 2.41, $p_{FDR} = 0.35$; child faces: $\beta_{age} = 0.0128$, t(125) = 2.06, $p_{FDR} = 0.41$, numbers: $\beta_{age} = 0.011$, t(125) = 2.16, $p_{FDR} = 0.41$). We found no evidence for other significant effects (Tables S5–6). Results were largely the same for a range of thresholds to define selective voxels (Fig. S4).

To further investigate how development within clustered category-selective regions contributes to the development of distinctiveness, we also conducted an analysis where we created independent disk-ROIs for each category-selective ROI (e.g., word-selective region) for each participant. We chose this approach to ensure that: (i) we examine information across voxels selective to one category, (ii) we use a constant number of included voxels across sessions of a participant, and (iii) that the voxel selection will not be biased to a particular session. Results show that distinctiveness develops for the category that was used to define the ROI (e.g., there is an increase in distinctiveness for words in the word-selective ROI), but also that development is not limited to the preferred category (Fig. S5).

## How does the nature of the internal representational space change from childhood to adolescence?

As there is a heterogeneous development of category distinctiveness both across categories and subsets of VTC voxels, it is interesting to consider how this relates to the neural representational space of these ten categories. To visualize the representational space and how it changes with age, we computed mean representational similarity matrices (RSMs) for 5–9-year-olds and 13-17-year-olds (Fig. S6A). We then used multidimensional scaling (MDS) to visualize the representational space in 2D. We focus on the development of the representational space in lateral VTC because it showed a more pronounced development compared to medial VTC; medial VTC data is in Figs. S7 and S8.

Visualization of the representational space of lateral VTC in children and teens illustrates two key findings.

First, distributed representations across both the union of selective voxels (Fig. 3a, d, Supplementary Movie 1) and non-selective voxels (Fig. 3b, e) have a categorical structure, however, the categorical representation over nonselective voxels is strongly diminished compared to that over the union of selective voxels (Fig. 3, compare A&D to B&E). That is, MDS embeddings over both the union of selective voxels (Fig. 3a, d) and over the non-selective voxels (Fig. 3b, e) reveal that representations of animate stimuli (faces, bodies) are largely separate from those of inanimate stimuli (objects, places, characters) in both children and adolescents. Yet, comparing the embeddings of the union of selective voxels (Fig. 3a, d) to that of the non-selective voxels (Fig. 3b, e) reveals that the representation of category information is much clearer and enhanced over the selective voxels.

Second, the representational structure over the union of selective voxels from childhood (small circles) to adolescence (large circles) reveals developmental changes. In contrast, the representational structure of the non-selective voxels remains largely unchanged. Examining the MDS embeddings over the union of the selective voxels reveals development of the categorical structure in several ways: (i) in both hemispheres, representations of faces strengthen and become more separable from other categories from childhood to the teens (Fig. 3a, d, red arrows moving outward, see developmental trajectory in Supplementary Movie 1), (ii) representations of both pseudowords and numbers strengthen from age 5 to age 17, and this development is particularly pronounced in the left hemisphere (Fig. 3a, d, blue arrows moving outward, Supplementary Movie 1), and (iii) representations of limbs in the right hemisphere weaken from childhood to the teens (Fig. 3d, Supplementary movie 1, yellow arrow moving inward). In contrast, comparing representations over the non-selective voxels in children and adolescents suggests no substantial development.

We quantified the development of the representational structure in both selective and non-selective voxels in each child (Fig. 3c, f, Methods). Thus, for each set of voxels we aligned the MDS embedding of each child's first session to that of their last session and measured the mean distance between the coordinates of each category in this shared embedded space. This analysis reveals that there is a significantly larger developmental change in the representation over the

union of selective voxels than the non-selective voxels in both hemispheres (left: t(28) = 6.40, $p < 0.001$, d = 1.19, t-test Fig. 3c, and right: t(28) = 4.92, $p < 0.001$, d = 0.91, Fig. 3f). Importantly, this effect is visible in most individual children illustrating within-child development of distributed responses (Fig. 3c, f). In sum, these analyses reveal that while there is categorical structure in both sets of voxels (union of selective, non-selective), the representation in the union of selective voxels (i) is enhanced compared to that of the non-selective voxels, and (ii) undergoes stronger development.

## Is development of distributed responses linked to improvements in behavior?

So far, accumulating evidence reveals the enhancement of distributed category representations in lateral VTC from age 5 to 17. Given that face recognition and reading also improve from age 5 to 17, we measured face recognition and reading ability in our longitudinal sample (Fig. S9) and tested if these developments are linked. We hypothesized that developmental increases in distinctiveness for faces and pseudowords may enhance recognition performance for these categories. We tested this hypothesis using LMMs relating behavioral performance to category distinctiveness, with participant as a random factor.

We found a significant and positive relationship between reading performance of pseudowords and distinctiveness for pseudowords over the union of the selective voxels of left lateral VTC (Fig. 4a, $\beta_{distinctiveness} = 40.19$ [95%-CI: 12.86;67.53], t(62) = 2.94, $p_{FDR} = 0.016$, LMM, random slope and intercept across participants). That is, better reading scores were associated with higher values of distinctiveness for pseudowords. The effect of distinctiveness predicting reading performance remained significant when age was added to the LMM ($\beta_{distinctiveness} = 28.92$ [4.82;53.02], t(61) = 2.4, $p_{FDR} = 0.046$; $\beta_{age} = 2.45$ [1.03;3.88], t(61) = 3.44, $p_{FDR} = 0.007$), showing that the effect of distinctiveness was independent from the effect of age. Importantly, this link between reading scores and distinctiveness for pseudowords in left lateral VTC was specific, as there was no evidence for a significant link between reading performance and (i) pseudoword distinctiveness over the non-selective voxels of left lateral VTC ($\beta_{distinctiveness} = 14.96$ [−8.14;38.05], t(62) = 1.29, $p_{FDR} = 0.35$), (ii) pseudoword distinctiveness over the selective voxels in right lateral VTC ($\beta_{distinctiveness} = -6.90$ [−34.5;20.8], t(62) = −0.50, $p_{FDR} = 0.72$), or (iii) face distinctiveness over the selective voxels in either hemisphere (left: $\beta_{distinctiveness} = 6.0$ [−17.42;29.44], t(62) = 0.51, $p_{FDR} = 0.72$; right: $\beta_{distinctiveness} = 0.72$ [−27.94;29.38], t(62) = 0.05, $p_{FDR} = 0.96$).

We also tested if a similar link exists between (i) the number of word-selective voxels in left lateral VTC or (ii) the size of the word-selective region pOTS-words in the left hemisphere and reading performance for pseudowords (Fig. S10AB). However, reading performance was neither significantly related to the number of selective voxels for pseudowords, nor the size of the left word-selective pOTS-words, suggesting that word distinctiveness in left lateral VTC is a better predictor of reading performance than the other metrics.

We further reasoned that if distinctiveness predicts behavior, then this model can be used to predict reading performance in new participants just from the distinctiveness of their lateral VTC selective voxels. We tested this prediction using a leave-one participant-out cross-validation (LOOCV) approach. Results reveal that distinctiveness for pseudowords in the union of selective voxels in left lateral VTC successfully predicts reading performance in left-out participants with a median prediction error of 8.7% (Fig. 4b, purple boxplot). Adding age to the model did not further reduce the prediction error (Fig. 3b, black boxplot; no evidence for a significant difference in performance, t(25) = 0.47, $p = 0.64$). In contrast, predicting reading performance from distinctiveness for pseudowords in non-selective voxels revealed a larger median prediction error of 11.5% (Fig. 4b, gray boxplot). We found a small but significant difference between the prediction error for the model using the distinctiveness over the selective voxels vs that

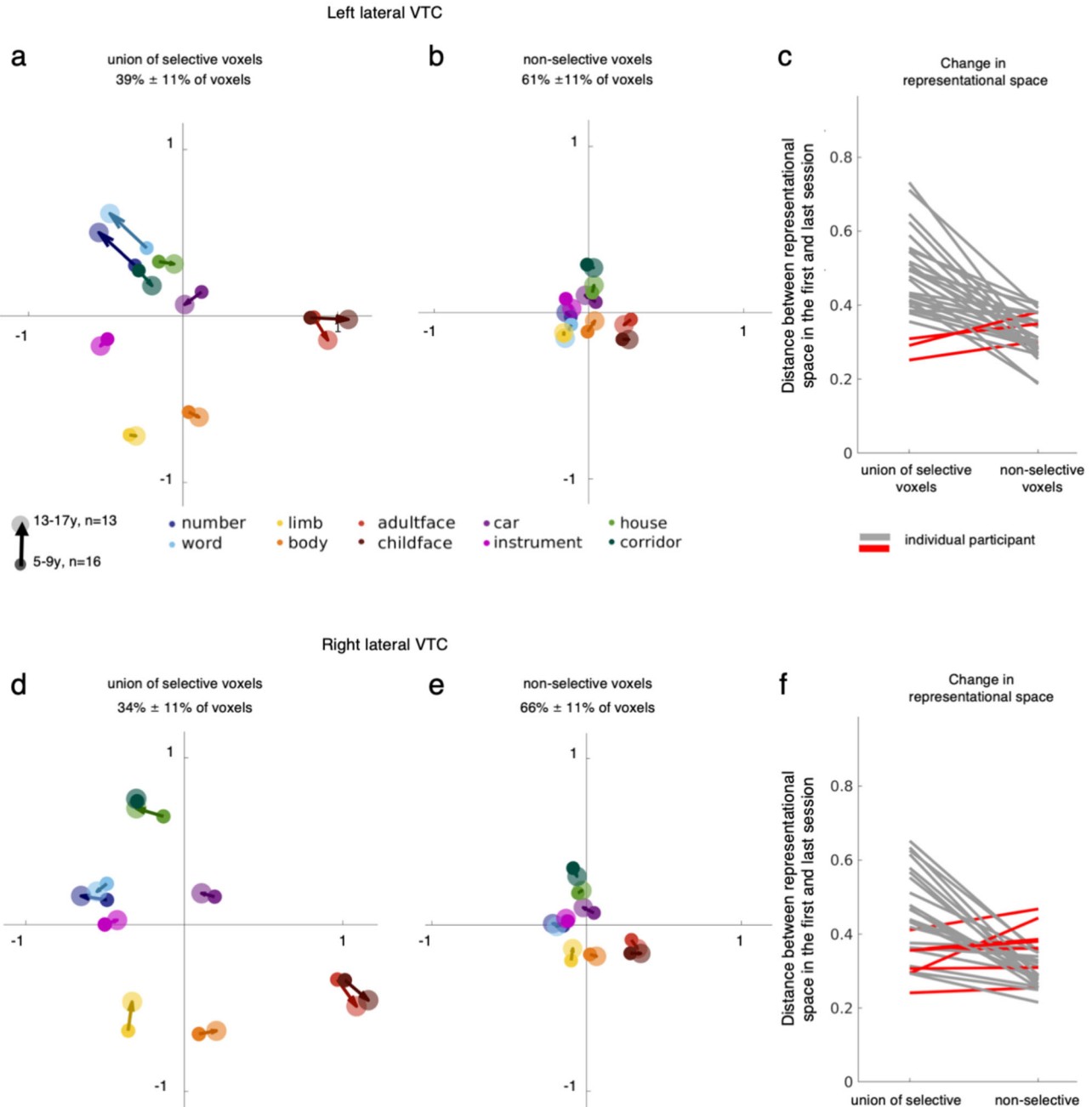

**Fig. 3 | Development of the representational space of the 10 categories in lateral VTC.** Multidimensional scaling (MDS) embeddings for the category representation in different subsets of voxels for two age groups: 5–9-year-olds ($n = 16$ participants, small circles) and 13–17-year-olds ($n = 13$ participants, larger circles). These age groups are used to illustrate the change in the representational space and are based on average RSMs of children in two age groups. All statistics are run using the full sample (Figs. 2, 3c, f). One session per child is included per MDS of each age group. **a**, **d** MDS embedding of the representational space across the union of selective voxels in left (**a**) and right (**d**) lateral VTC. **b**, **e** MDS embedding the representational space of the remainder, non-selective voxels of left (**b**) and right (**e**) lateral VTC. **c**, **f** Line plots depicting the change in representational spaces in individual children in left (**c**) and right (**f**) lateral VTC across the selective and non-selective voxels. The change in representation is the mean Euclidian distance between category positions in the MDS embedding of a child's first session vs their last session. Each line is a participant ($n = 29$); *Gray*: larger distances in the selective voxels; *Red*: larger distances in the non-selective voxels.

over the non-selective voxels (Fig. 4b-swarm plot, two-sided t-test comparing the difference in error to zero: $t(25) = -2.11$, $p = 0.045$, $d = 0.41$). That is, for most participants the prediction error was larger for a model based on distinctiveness of the non-selective voxels than a model based on distinctiveness of the selective voxels.

Likewise, distinctiveness for adult faces in the union of selective voxels of right lateral VTC was significantly and positively related to face recognition performance ($\beta_{distinctiveness} = 31.49$ [15.07;47.92],

$t(80) = 3.82$, $p_{FDR} = 0.0008$, LMM with random slope and random intercept across participants). That is, better face recognition performance was associated with higher values of distinctiveness for faces (Fig. 4c). When age was added to the LMM the effect of distinctiveness was only trending and was no longer significant after correction for multiple comparisons ($\beta_{distinctiveness} = 16.41$ [−0.19;33.01], $t(79) = 1.97$, $p_{FDR} = 0.08$; $\beta_{age} = 3.94$ [2.94;4.93], $t(79) = 7.89$, $p_{FDR} < 0.001$). Nonetheless, the link between face recognition and face distinctiveness was

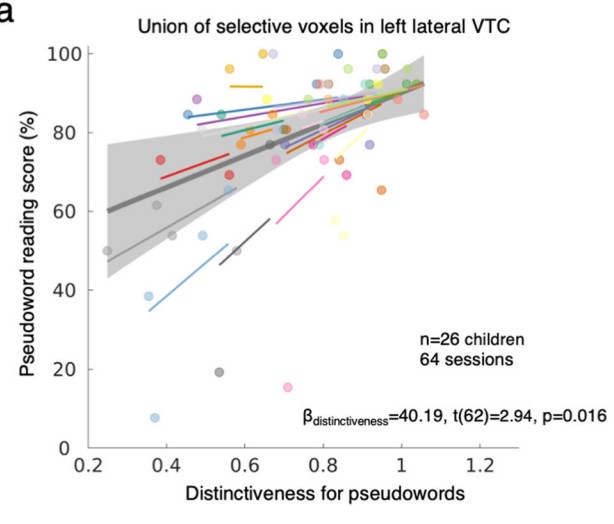

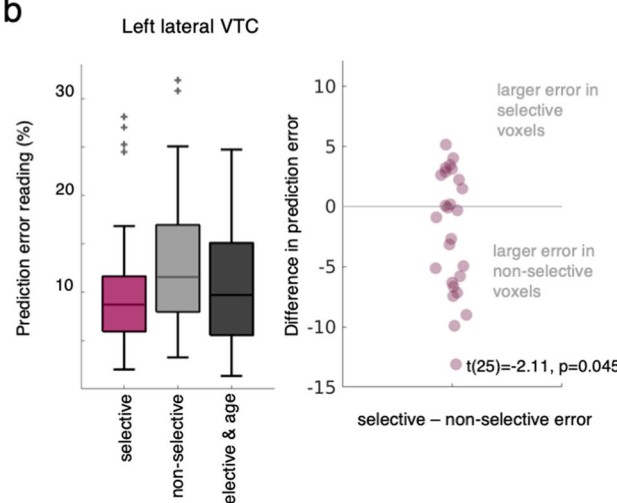

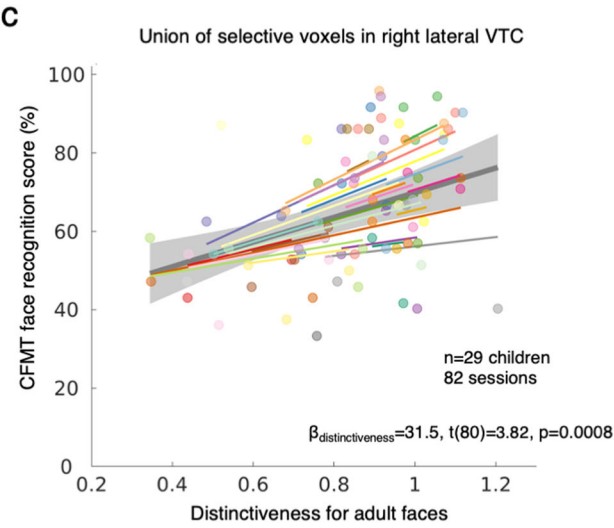

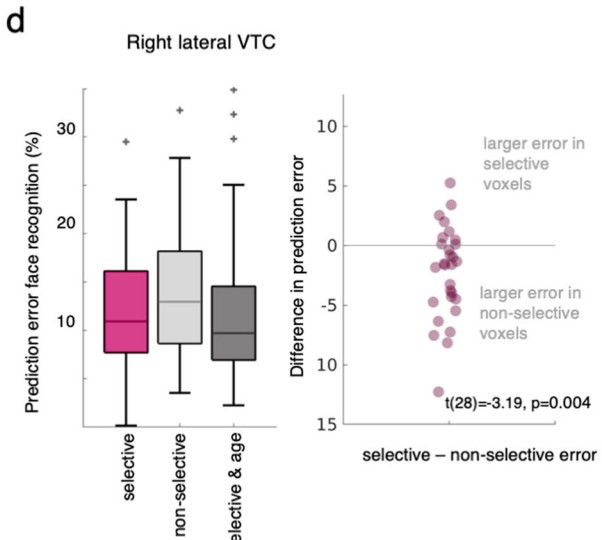

**Fig. 4 | Distinctiveness for words and faces in left and right lateral VTC, respectively, predict reading and face recognition performance in individual children. a** Linear mixed model (LMM) with random slopes and intercepts relating reading performance of pseudowords (Woodcock Reading Mastery Test, WRMT) to pseudoword distinctiveness over the union of selective voxels of left lateral VTC. Model parameters indicated in the bottom, p-Value adjusted for multiple comparisons (see text). Each dot is a session; Dots are colored by participant. *Colored lines*: individual slopes and intercepts. *Thick gray line*: LMM prediction. *Shaded gray*: 95% CI of the slope. **b** *Left*: Median error in predicting the reading performance of a left-out participant from their distinctiveness for pseudowords in left lateral VTC using the parameters derived from the LMM of the rest of the participants (leave-one-out-cross validation). Higher values indicate worse model prediction. The box plots show the prediction error for three different models: *Selective*: LMM predicting behavior from distinctiveness over the union of selective voxels. *Non-selective voxels*: LMM predicting behavior from distinctiveness over the non-selective voxels. *Selective & age*: LMM predicting behavior from distinctiveness over the union of selective voxels with age as an additional factor. Boxplots show the 75% and 25% percentiles (shaded areas) and the median (horizontal lines). Whiskers extend to the most extreme data points not considered outliers (values more than 1.5 times the interquartile range away from the bottom or top of the box). *Gray plus signs*: outliers. *Right*: Swarm plots showing the difference between the prediction error for selective vs. non-selective voxels. Each dot is a participant. Statistics of the two-sided t-test (at bottom), *n* = 26. **c** Same as (**a**) but for LMM parameters for a model relating face recognition performance (Cambridge face recognition memory test (CFMT), adult faces) and distinctiveness for adult faces over the selective voxels of right lateral VTC. **d** Same as (**b**) but for face recognition performance and distinctiveness for adult faces in right lateral VTC, *n* = 29.

specific to the selective voxels of right lateral VTC, as no evidence was found for a link between (i) face recognition and face distinctiveness in the non-selective voxels of right lateral VTC ($\beta_{distinctiveness} = 0.85$ [−18.14;19.84], t(80) = 0.09, $p_{FDR} = 0.93$), (ii) face recognition and face distinctiveness in the union of the selective voxels of left lateral VTC ($\beta_{distinctiveness} = 17.79$ [−0.16;35.73], t(80) = 1.97, $p_{FDR} = 0.08$) or (iii) face recognition and pseudoword distinctiveness in the selective voxels in right lateral VTC ($\beta_{distinctiveness} = -8.7$ [−23.21;5.91], t(80) = −1.18, $p_{FDR} = 0.31$). While face recognition ability was also linked to

distinctiveness for pseudowords in the left hemisphere ($\beta_{distinctiveness} = 24.56$ [6.77;42.34], t(80) = 2.75, $p_{FDR} = 0.017$), this link was driven by age, as there was no evidence for a significant relationship when age was added to the model ($\beta_{distinctiveness} = -2.69$ [−17.65;12.28], t(79) = −0.36, $p_{FDR} = 0.81$, $\beta_{age} = 4.18$ [3.12;5.24], t(79) = 7.83, $p_{FDR} < 0.001$). As prior research has also suggested a link between face recognition performance and the number of face-selective voxels[21] we also examined this relation in our data. We found that face recognition was also significantly linked to the number of

face-selective voxels and the size of the right face-selective pFus-faces (Fig. S10C, D), but once age was added to the model there was no evidence for these effects.

Notably, distinctiveness for adult faces over the selective voxels of right VTC predicts face recognition performance in left out participants (LOOCV, Fig. 4d-purple boxplot). Face recognition ability is predicted from face distinctiveness over the selective voxels of right lateral VTC in the left-out participant with a median prediction error of 10.93%. There was no evidence for a smaller error when age was added to the model (t(28) = −0.42, p = 0.68, Fig. 4d-dark gray boxplot) and for most participants, the prediction error was higher from a model based on the non-selective voxels than a model based on the union of selective voxels (Fig. 4d-light gray boxplot and swarm plot, t-test comparing the difference in error to zero: t(28) = −3.19, p = 0.004, d = 0.59).

Together these data suggest that the distinctiveness of word and face representations in lateral VTC predicts performance for these categories, and that developmental improvements in reading and face recognition abilities are linked to increases in word and face distinctiveness in the selective voxels of left and right lateral VTC, respectively.

## Discussion

By combining longitudinal measurements of distributed VTC responses to multiple categories with behavioral data in children over several years, the present study reveals several findings regarding the functional development of category representations in children's brains. First, our data reveals the development of multiple category representations in VTC from age 5 to age 17. This development consists of both enhancement and diminution of the distinctiveness of distributed category representations, as well as the development of representations for categories that are not associated with clustered regions in this cortical expanse. Second, the development of distributed responses was driven mainly by changes in representations over the union of voxels that exhibited category selectivity. Third, importantly, the developmental increases of category distinctiveness to words and faces in the union of selective voxels in the dominant hemisphere (left and right lateral VTC, respectively) predicted the visual recognition ability of words and faces in left-out participants.

Our longitudinal and multivariate analyses of category distinctiveness in lateral VTC reveal increases of distinctiveness for faces, words, numbers, and houses as well as a decrease in distinctiveness for limbs. In addition, in medial VTC our analyses reveal increases in distinctiveness for houses and faces. The increase in distinctiveness for faces and words in left lateral VTC is in line with prior findings showing that word- and face-selective regions grow with development and become increasingly more selective to their preferred category[16,19,21,23,25,32] and with prior findings using multivariate approaches showing developmental increases in distinctiveness for words[16,18]. Likewise, the increase in distinctiveness for houses in medial VTC is in line with findings showing childhood development of the place-selective region in medial VTC[21,33]. Further, the decrease in distinctiveness in distributed lateral VTC responses for limbs is consistent with findings that limb-selective regions shrink from childhood to the teens and lose their limb-selectivity[23]. Together these data are consistent with the idea that developmental changes in the degree of selectivity affect distributed responses. That is, these data suggest that developmental increases in category selectivity lead to increases in category distinctiveness, and developmental decreases in selectivity lead to decreases in distinctiveness.

Our data also reveal increases in distinctiveness for numbers and houses with age in the right lateral VTC and for faces in the medial VTC. The development of distinctiveness for numbers is in line with the expectation that schooling not only enhances word but also number representations in VTC. As prior cross-sectional[18] and univariate[23]

approaches did not detect the development of number responses in lateral VTC, these data highlight the higher sensitivity of longitudinal, multivariate approaches to measuring functional development in the brain. Given that place-selective regions are located in the medial part of VTC[10,12], finding developmental increases in distinctiveness for houses in lateral VTC shows that increases in distinctiveness can also occur in parts of VTC that don't have a clustered region for that category. In line with this idea, we also observed an increase in distinctiveness for faces in medial VTC, even as it does not contain a clustered region selective to faces. As both voxels with strong positive and negative preferences can contribute to category information (e.g., words[18]), the developmental increases in distinctiveness may be driven both by increases in the number of voxels selective to the respective category, as well as increases in the number of voxels with a negative preference to that category. For instance, in the case of distinctiveness for houses in lateral VTC, it is possible that this increase was driven by both increases in the number of house-selective voxels[23] and increases in the number of voxels with a stronger negative preference to houses, such as face-selective voxels.

The present data also offers insight on the laterality of the development of category distinctiveness[34]. In particular, prior research has suggested a graded developmental lateralization of word representations to the left hemisphere[11,18,31,35] and of face representations to the right hemisphere[7,31,35]. Consistent with this hypothesis, we see that the development of distinctiveness for pseudowords was significant only in the left hemisphere and that distinctiveness in the left, but not right hemisphere, predicts reading performance. With regard to the lateralization of face representations, our results reveal a bilateral development of distinctiveness across all voxels in lateral VTC, as well as across the union of category-selective voxels, in line with prior studies finding that the number of face-selective voxels increases across development in both hemispheres[23]. It should be highlighted at this point that the results shown in Figs. 1 and 2 summarize the change in distinctiveness with age, rather than the level of face distinctiveness. That is, while the change in face distinctiveness was bilateral, distinctiveness for faces was nonetheless numerically higher in the right than left hemisphere. As such, these results support lateralization of word and face representations in children and teens but suggest that development is not restricted to the dominant hemisphere.

In addition to the changes in category distinctiveness, we also examined changes in the nature of the representational space of the 10 categories across VTC. Like prior cross-sectional studies[15,17] we find that in lateral VTC representations of faces are already separated from those of other categories at age 5 (Fig. 3) and yet our data show that they continue to be enhanced into adolescence. Our results also show that representations for words and numbers in the left hemisphere are not yet separated from those of other categories in the younger age group of 5-9-year-olds but become enhanced by adolescence. As such, these data suggest (i) that the development of distributed face representations starts earlier than that of character representations, which appears to start later in childhood, and (ii) that schooling and learning how to read changes category representational space in VTC. We note that other extensive visual experience during childhood (e.g., playing Pokémon) also affects category representations in VTC[36].

In addition to the enhancement of character and face representations during childhood, our data also reveal that representations for limbs weakened from childhood to adolescence in the right hemisphere in lateral VTC. That is, representations for limbs move towards the center of the MDS space as children get older (Fig. 3d). While the reason for this developmental change is currently unclear, it is possible that changes in visual experience with limbs during childhood contribute to this change. Future longitudinal studies can test this prediction empirically. These findings also raise questions on how the representational space for visual categories may be affected in individuals with altered or deprived visual experience such as children

with cataracts[37,38], illiterate individuals, or late-literate individuals who only learned how to read when they were adults[39]. These results also highlight the importance of examining the development of category representations in other developmental phases, in particular during infancy[40–43] and testing how these developments relate to other behaviors, such as eye gaze patterns toward visual categories[44].

The present results also have important theoretical implications for the debate on how distributed[13] vs. modular[28] category representations in VTC contribute to behavior. On the one hand, our data reveal that representations of animate (faces and bodies) categories are separable from representations of the other inanimate categories in both selective and non-selective voxels (Fig. 3), consistent with the predictions of the distributed hypothesis[13]. On the other hand, we find significant development of category distinctiveness across the union of the selective voxels in lateral VTC, but no evidence for the development of category distinctiveness in the non-selective voxels, contrary to the predictions of the distributed hypothesis[13]. Similarly, the analysis of the development of distinctiveness in disk ROIs that are selective to one particular category (Fig. S5) showed that distinctiveness developed for the category used to define the ROI (i.e., distinctiveness for words developed in the word-selective ROI) but also that development was not limited to the ROI-defining category. Interestingly, the decrease in distinctiveness for limbs that was observed across all voxels in lateral VTC, was not significant when examining the union of the selective voxels or the non-selective voxels as separate subsets. This suggests that both subsets of voxels may contribute to the decrease in distinctiveness for limbs.

Crucially, however, we find that distinctiveness over the union of selective voxels, but not the non-selective voxels, predicts recognition behavior. That is, distinctiveness for pseudowords over the union of selective voxels in left lateral VTC predicted pseudoword reading with ~9% error in left-out participants; likewise, distinctiveness for faces over the union of selective voxels in right lateral VTC predicted face recognition with a ~11% error in left out participants. These longitudinal measurements are consistent with our prior cross-sectional data that found that better reading ability in adults than children is coupled with higher distinctiveness in word-selective voxels[18] but also show that the link between reading and distinctiveness for words in the union of the selective voxels of left lateral VTC was higher than for face recognition and face distinctiveness in the union of selective voxels of right VTC. In particular, the link between face recognition performance and distinctiveness for faces over the union of selective voxels in the right hemisphere was no longer significant when age was added to the model. We hypothesize that the more restricted range of face recognition than reading performance combined with smaller individual variability in the rate of face recognition performance changes over time may contribute to these differences. Future longitudinal studies that include participants with both a larger range of ages and a larger range of face recognition abilities (comparable to the range of reading abilities) will be important to tease apart these possibilities. Together, the present data support a sparsely distributed account of VTC functional organization[29]. That is, while we find that distributed responses across both selective and non-selective voxels contain category information, the development of the selective voxels is more enhanced and better predicts behavioral outcomes.

The described link between category distinctiveness in the union of selective voxels and recognition ability has important ramifications for studying the neural basis of atypical development and developmental disorders. For instance, future research can examine if children with dyslexia have lower distinctiveness for words in left lateral VTC than age-matched typically-developing children and whether distinctiveness for words contributes to explaining reading ability independent of other predictors such as phonological awareness[45], socioeconomic status[46,47], white matter properties[48], gyrification in auditory cortex[49], or perceptual decision-making[50]. Similarly, future research can test if adults and children with developmental prosopagnosia[51,52] show lower distinctiveness for faces, compared to typical age-matched controls, and conversely, if super-face-recognizers[53,54], show higher distinctiveness for faces in right lateral VTC than typical age-matched controls.

In sum, our results not only provide insights on the development of category representations in VTC, but also elucidate how this development relates to behavioral changes in visual recognition of faces and written words during childhood and adolescence. Future studies can test the hypothesis that this link between category distinctiveness and behavior generalizes to the representation of other categories and skills involving visual recognition, such as number representations and math.

## Methods

### Participants

Children aged 5–12 years with normal or corrected-to-normal vision were recruited for this study. This age range was selected for two reasons: First, face recognition and reading, the two behavioral measures assessed in this study, improve during this age range. Second, prior studies investigating the functional development of VTC have shown development in this age range[16,23].

Children were recruited from schools in and around Palo Alto, CA. The diversity of the participants reflects the makeup of the region: 62.5% of children were Caucasian, 20% were Asian, 5% were Native Hawaiian, 5% were Hispanic, and 7.5% were multiracial or from other racial/ethnic groups. We collected fMRI data from 40 (26 female) children (onset age = 5–12 years, M = 8.66 years, SD = 2.34 years). Data from 4 children had to be excluded because they dropped out of the study after participating only once, thus providing no longitudinal fMRI data. Data from 7 children were excluded because their data did not pass the inclusion criteria (see below). In the remaining 29 children, 29 functional sessions were excluded due to motion, 1 session due to a technical error during acquisition, and 1 session due to aliasing artifacts during acquisition. The fMRI data has previously been reported in[23].

Therefore, in this study, we report data of 29 neurotypical children (18 female, 11 male), who were between 5 to 12 years old (mean = 9.19, SD = 2.13) when they enrolled in the study. Sex and gender were not considered in the design of the current study. Gender was determined based on self-reporting. No sex and gender-based analyses were performed as the final sample was not completely balanced according to these factors and the limited sample size does not enable deriving meaningful conclusions regarding sex and gender effects. Children participated in the study for 1 to 5 years. When possible, children completed 1 to 2 functional scans and a structural scan each year. Each child participated in at least 2 and up to 10 fMRI sessions (mean = 4.41, SD = 1.92) with the time interval between the first and last fMRI scan ranging from 10 months to 5 years (mean = 45 months, SD = 18 months). The sample size in the present study is similar to those reported in prior cross-sectional publications[55,56] and larger than previous longitudinal studies on VTC development[16]. No statistical method was used to predetermine the sample size. We also report behavioral data collected on a subset of sessions of the same children (see below, behavioral data collection).

### Statement on ethical regulations

This study was approved by the Institutional Review Board of Stanford University and complies with all relevant ethical regulations. Prior to the start of the study, parents gave written consent, and children gave written assent for their participation. Children received $30 per hour for scanning, $10 per hour for behavior, and a small toy for their participation.

## Procedure

Before taking part in the actual fMRI session, children completed training with a mock scanner to enhance the quality of pediatric neuroimaging data. During the mock scanner training children were acclimated to the scanner environment in a child-friendly way: Children practiced laying still while watching a short movie and receiving live feedback on how much they were moving. After the mock scanner training the child participated in the actual MRI session. Functional and anatomical scans were typically conducted on different days to avoid fatigue. Face recognition and reading tests were typically completed after one of the scanning sessions.

## Magnetic resonance imaging

**Structural imaging.** We collected neuroimaging data at the Center for Cognitive Neurobiological Imaging at Stanford University using a phase-array 32 channel head coil and a 3 Tesla GE Discovery MR750 scanner (GE Medical Systems). Anatomical scans were collected using quantitative MRI (qMRI[57]). Here, we used a spoiled gradient echo sequence with multiple flip angles ($\alpha = 4°, 10°, 20°, 30°$), TR = 14 ms and TE = 2.4 ms. The scan resolution was $0.8 \times 0.8 \times 1.0$ mm$^3$ (which was later resampled to 1 mm isotropic). For T1-calibration spin-echo inversion recovery scans were acquired with an echo-planar imaging read-out, spectral spatial fat suppression and a slab inversion pulse. These scans were collected at TR = 3 s, inplane resolution = 2 mm x 2mm, slice thickness = 4 mm and 2x acceleration, echo time=minimum full.

**Functional imaging.** The functional data was acquired with the same scanner and head coil as the structural images. We oriented slices parallel to the parieto-occipital sulcus. Data were collected using a simultaneous multi-slice, one-shot T2* sensitive gradient echo EPI sequence with a multiplexing factor of 3. This sequence had a FOV = 192 mm, TR = 1 s, TE = 30 ms, and flip angle = 76°, a resolution of 2.4 mm isotropic and near whole brain coverage (48 slices).

## 10 category experiment

During functional imaging, children completed three runs of a 10-category experiment[23,58]. In each functional run, participants watched images from 10 categories which can be grouped into 5 domains (Supplementary Fig. 1B): faces (child faces, adult faces), characters (pseudowords, numbers), body parts (headless bodies, limbs), places (houses, corridors) and objects (cars, string instruments). The authors affirm that the individuals displayed in Supplementary Fig. 1B provided written informed consent for the publication of these images. Images contained category stimuli, which were placed on a phase-scrambled background generated from randomly selected images. All stimuli were grayscale. Images were presented at a rate of 2 Hz and did not repeat across the course of the experiment. Images were presented in 4 s blocks, which were intermixed with baseline blocks showing a gray luminance screen. Blocks were counterbalanced across categories and baseline blocks. Participants were instructed to view the images while fixating on a central dot and perform an oddball task. The oddball task required participants to press a button whenever an image containing only the phase-scrambled background appeared. Behavioral responses to the oddball task were recorded in 98 out of 128 sessions due to occasional button malfunction. Performance on the oddball task performed during scanning was overall high (median performance = 91%, SD = 18%).

## Behavioral data collection

**Assessing reading ability.** In a subset of sessions participants also completed the word identification and word attack tests from the Woodcock Reading Mastery Test (WRMT). Reading assessment was performed outside the scanner. In the word identification task, participants are asked to read a list of words as accurately as possible. In the word attack task, participants are supposed to read a list of pseudowords as accurately as possible. Tests do not have a time limit, but end when the participant makes 4 consecutive errors or has read the complete list. The reading score for each test was obtained by dividing the number of words read correctly by the total number of words and multiplying the result by 100.

**Assessing face recognition ability.** In a subset of sessions participants also completed outside the scanner the Cambridge Face Memory Test (CFMT[59]) using adult male faces and a version of the CFMT using male child faces[60]. The CFMT is a self-paced face recognition test. In the learning phases, participants learn the identity of six target unfamiliar faces. In the test phase, in each trial they are shown a triplet of faces and are asked to identify the learned faces amongst distractor faces. The test constitutes 72 trials that become increasingly difficult as faces appear in unknown views, lighting, and superimposed noise. Accuracy was measured as the percent of correct responses made during the test phase.

## Data analysis

We used MATLAB version 2017b (The MathWorks, Inc.) and the mrVista software package (version 2.1, https://github.com/vistalab/vistasoft/wiki/mrVista) to analyze the data. Swarm plots in Fig. 4 were created using MATLAB version 2020b.

**Inclusion criteria.** There were two criteria relevant for the inclusion of the fMRI data in the analysis. First, there needed to be at least 2 (out of 3) runs per session with within-run motion <2 voxels and between-run motion <3 voxels. Second, the child participated in at least two fMRI sessions that were at least 6 months apart. Because in several sessions only 2 out of 3 functional runs passed the motion quality criteria, final analyses include 2 runs per session to ensure equal amounts of data across participants and functional sessions. For sessions with 3 runs surviving motion quality thresholds, the 2 runs with the lowest within-run motion were selected.

For analyses that relate fMRI data to behavioral data, behavioral datasets (face recognition and reading tests) were included in the analysis if the time between acquisition of the behavioral data and the acquisition of fMRI data was <1 year (Fig. S1A).

**Analysis of structural MRI data and individual template creation.** We processed quantitative whole-brain images of each child and timepoint with the mrQ pipeline (https://github.com/mezera/mrQ[57]) to generate synthetic T1 brain volumes. For each child, we then used the synthetic T1 brain volumes from their multiple timepoints to generate a within-participant brain volume template. The individual template of each child was generated using the FreeSurfer Longitudinal pipeline implemented in FreeSurfer version 6.0. (https://freesurfer.net/fswiki/LongitudinalProcessing[61]). We then manually edited the gray-white matter segmentations of each participant's within-participant brain template to fix segmentation errors (like holes and handles) to generate an accurate cortical surface reconstruction. The functional data from each child's multiple timepoints (see below) was then aligned to the within-participant template. The reasons for this procedure were to: (i) minimize potential biases which can occur from aligning longitudinal data to the anatomical volume from a single timepoint[61] and to (ii) enable the use of the same anatomical regions of interest (ROIs) across different timepoints in the same brain volume for each participant. On average 2.48 (SD = 0.69) synthetic T1s were used to generate the within-participant-template (min = 2, max=5). In 17 participants, the last fMRI session that was included was conducted after the within-participant template had been created. These functional sessions were acquired on average 11 ± 2 months after acquisition of the last synthetic T1 that was included in the within-participant-template (excluding 2 participants whose last synthetic T1 could not be

used because of technical error during acquisition and participant motion).

**Definition of lateral and medial VTC Regions of Interest (ROIs).** We defined lateral and medial VTC ROIs based on anatomical landmarks on the inflated cortical surface in each hemisphere of each participant as in prior publications[18,23]. The posterior border of both VTC ROIs was defined along the posterior transverse collateral sulcus (ptCoS). The anterior border was aligned with the posterior end of the hippocampus, which typically aligns with the anterior tip of the mid-fusiform sulcus (MFS). The lateral border of the lateral VTC ROI was placed along the inferior temporal gyrus (ITG). Lateral and medial VTC ROIs were separated by placing a border along the mid-fusiform sulcus (MFS). The medial border of the medial VTC ROI was placed along the Collateral Sulcus (CoS).

**Analysis of functional MRI data.** Functional data from each session were aligned to the individual participant's brain template (see above). Data was motion-corrected both within and across functional runs. We applied no spatial smoothing and no slice-timing correction. Time courses were transformed into percentage signal change. To this end, each timepoint of each voxel's data was divided by the average response across the entire run. A general linear model (GLM) was fit to each voxel by convolving the stimulus presentation design with the hemodynamic response function (as implemented in SPM, https://www.fil.ion.ucl.ac.uk/spm/) to estimate the contribution of each of the 10 conditions. Analyses were performed in voxel space (not on surface nodes).

**Multivariate pattern analysis.** For each category multivoxel patterns (MVPs) of responses are vectors of response amplitudes across all voxels in each ROI (left and right lateral and medial VTC). Amplitudes were estimated from the GLM run at each voxel. Response amplitudes were transformed into z-scores to remove between voxels differences in amplitudes (e.g., because of distance from the coil) and to down-weight noisy voxels (see[18,62,63]). Next, we calculated all pair-wise correlations between MVP pairs from one functional run to the other resulting in a $10 \times 10$ representational similarity matrix (RSM[14]) for each session.

**Assessment of category distinctiveness.** For each session's RSM, we computed the distinctiveness for each of the 10 categories. The distinctiveness of a category is defined as the within-category minus between-category similarity of distributed responses leaving out the between-category similarity with the other category from the same domain. This is illustrated in the gray box in Fig. 1a: In the RSM the values on the diagonal represent the within-category similarity, while the off-diagonal values represent the between-category similarities. For instance, to compute the distinctiveness of words, the between-category similarities for all categories except numbers are subtracted from the within category similarity for words. The reason for leaving out the other category from the same domain is that for some domains the two categories are very similar to each other (such as adult and child faces) while for other domains the two categories differ from each other to a greater extent (such as cars and string instruments) so we chose a procedure that would be least biased and will not differently affect stimuli from various domains.

**Definition of voxel subsets: the union of selective voxels across all categories and of non-selective voxels.** To define the union of selective voxels in each session, we first computed the selective voxels for each category and then took the union of selective voxels across all 10 categories. Selective voxels for each category were defined by contrasting responses to a given category vs. all other categories except the other category from the same domain (i.e., words vs. all

other categories except numbers). Category-selectivity was defined as a $t$-value $> 3$ (voxel-level) for the contrast of interest. This threshold was chosen because prior research has shown that category-selective regions can be defined reliably in individual participants including both children and adults using this threshold[23,31,55,56,64]. Then, we took the union of the selective voxels across all 10 categories. Non-selective voxels were the remainder of voxels in lateral VTC, that were not selective of any of the 10 categories. For the control analyses shown in Figs. S3 and S4 we defined the two voxel subsets for $t$-values $> 1$, $t$-values $> 2$, $t$-values $> 3$ (as presented in the main analysis), $t$-values $> 4$, and $t$-values $> 5$.

**Control analysis matching the voxel subsets for variance explained.** To test if the lack of development in the non-selective voxels (Fig. 2a) was driven by poor-responding voxels, rather than well-responding voxels without any known category selectivity, we performed a control analysis using subsets of the union of the selective voxels and of the non-selective voxels in lateral VTC that were matched on variance explained. For this purpose, 300 voxels were chosen in each session and each voxel subset. This number was chosen (i) to select the same number of voxels across subsets, and (ii) to ensure that the analysis can be conducted in all sessions, including sessions with a low number of selective voxels. For the non-selective voxels, the voxels with highest variance explained were selected, for the union of selective voxels a random selection of voxels was performed. Next, we repeated the analyses shown in Fig. 2 using these voxel subsets (Fig. S2).

**Control analysis in category-selective disk ROIs.** The goal of this analysis was to (i) determine changes in distinctiveness in ROIs that show selectivity to a given category (i.e., in a word-selective ROI) and (ii) to determine changes in distinctiveness in voxel sets that are constant across sessions of a given participant (Fig. S5). For this analysis disk-ROIs with a radius of 10 mm were created for each category-selective ROI for each participant. Disk ROIs were created for the word-selective ROIs pOTS-words and mOTS-words (also called the Visual Word Form Area[11]), the face-selective ROIs pFus-faces and mFus-faces (also called the Fusiform Face Area[7]), the limb-selective ROI OTS-limbs (also called the Fusiform Body Area[8]), and the place-selective ROI CoS-places (also called the Parahippocampal Place Area[10]). For the word ROIs, only the left hemisphere was included because the ROIs could only be defined in the right hemisphere in about 20% of participants[23].

To define the disk ROIs, ROIs of all sessions of that contrast in a given participant (i.e., word-selective ROI in session 1, 2 and 3) were used, which had individually been defined in the respective participant's functional sessions[23]. Then, the average center coordinates of these ROIs across all sessions of that participant were computed and the 10 mm disk ROI was centered on those average coordinates. This approach was chosen as (i) it ensures that the voxels will show selectivity to a certain category, (ii) it ensures that the number of included voxels is constant across sessions of a participant, (iii) provides an independent definition of the ROI, and (iv) minimizes bias towards a particular session.

**Visualizing changes in the representational space.** We used multi-dimensional scaling (MDS) to visualize the representational space in lateral and medial VTC and determine how it changes with age (Fig. 3a, b, d, e & Supplementary Movie 1, Supplementary Fig. S8). As visualizing how the representational space changes longitudinally is challenging, we show MDS embeddings for young children (5–9-year-olds) and teenagers (13–17-year-olds). Supplementary Movie 1 is a video that shows the developmental trajectory of the MDS embedding in lateral VTC across age. These age groups are used for visualization only; statistics are run on the whole longitudinal sample (see below). For the MDS embedding shown in Fig. 3a, b, d, e we first computed mean RSMs

for 5–9-year-olds and 13–17-year-olds separately for the union of selective voxels and for the non-selective voxels in each hemisphere (Fig. S6C, D). The RSMs were then turned into dissimilarity matrices. We next applied classical multidimensional scaling (MDS) to visualize the representational space in 2D (Fig. 3a, b, d, e). We used the same approach to visualize the development of the representational space in medial VTC (Supplementary Figs. 7, 8). To facilitate visual comparison of different MDS embeddings across hemispheres and sets of voxels, we aligned all MDS embeddings of 13–17-year-olds to the embedding of 13–17-year-olds across all voxels in left lateral VTC using the Procrustes transformation. We next used the Procrustes transformation to align the embedding of 5–9-year-olds to that of 13–17-year-olds. Procrustes analysis was used without scaling in both cases.

**Assessing changes in the representational space.** The line plots in Fig. 3c, f quantify the change that is visualized in the MDS embeddings (Fig. 3a, b, d, e). The goal of this analysis was to assess how the representational structure changes in the union of selective voxels compared to in the non-selective voxels in each child. To this end, we applied MDS to the RSM of each child's first session in the study and to the RSM of their last session. We next used Procrustes transformation without scaling to align the embedding of the first session to that of the last session. In each child we then measured the Euclidian distance for each category between the first to the last session, in the shared MDS embedding space and report the mean distance across all categories. This analysis was performed for both hemispheres and for both sets of voxels (union of selective voxels, non-selective voxels).

**Statistics**

Unless stated otherwise, tests reported in this manuscript are two-tailed.

We used linear mixed models (LMM) to test whether distinctiveness develops with age because LMMs can account both for the hierarchical data structure with multiple sessions being nested within each participant and for the uneven distribution of sessions across time (Supplementary Fig. 1A). Normality and equal variances were not formally tested.

**Statistical analyses related to Fig. 1.** LMMs were fitted using the '*fitlme*' function in MATLAB version 2017b (The MathWorks, Inc.). In these models, category distinctiveness was predicted by age using participant as a random factor. We first tested whether a random intercept or random slopes model fit the data best. In a random intercept model, intercepts are allowed to vary across participants, while in a random slope model both intercepts and slopes can vary across participants. Since in the majority of cases random intercept models fit the data best, we used random intercepts for all analyses presented in Fig. 1 to enable comparability across models.

We next tested whether motion during scanning and tSNR contributed to the model fit. Adding motion as a predictor to the LMM did not significantly contribute to the model fit except for distinctiveness for string instruments. Importantly, adding motion in the model for distinctiveness for string instruments did not influence the result: There was no significant contribution of age to string instruments distinctiveness with or without adding motion. We next tested whether tSNR contributed to the model fit. Since adding tSNR contributed to the model fit independently from age in several cases, we included tSNR as an additional factor into the model. The bars in Fig. 1d show the slopes for the effect of age on category distinctiveness taking into account tSNR.

LMMs related to Fig. 1d, e can be expressed as: category distinctiveness ~ age in years + tSNR + (1|participant), in which category distinctiveness is the response variable, age and tSNR are the predictors and the term (1|participant) indicates that intercepts vary by participant.

False-discovery rate (FDR) correction following the procedure by Benjamini and Hochberg[65] as implemented in MATLAB version 2017b (The MathWorks, Inc.) using the *mafdr* function was used to account for multiple comparisons (20 tests) for analyses shown in Fig. 1d, e. This function computes false-discovery-rate-adjusted p-values based on a ranking of *p*-values. Adjusted *p*-values are reported in the text.

**Statistical analyses related to Fig. 2.** The analyses in Fig. 2 were performed similar to those in Fig. 1d, e and can be expressed as: category distinctiveness ~ age in years + tSNR + (1|participant).

LMMs were run separately for each category, hemisphere, and voxel subsets (union of the selective voxels, non-selective voxels). Here, *tSNR* was obtained for each voxel and then averaged across voxels within each voxel subset (union of the selective voxels, non-selective voxels). In addition, we ran a LMM to test whether there are differences in *tSNR* in the subset of the union of the selective voxels and the non-selective voxels.

This model can be expressed as: tSNR ~ voxelSubset + (1|participant), where voxelSubset is a binary predictor (selective/non-selective).

For analyses shown in Fig. 2 FDR correction was performed as described for Fig. 1, separately for analyses in lateral and medial VTC.

**Statistical analyses related to Fig. 3.** We used paired t-tests to evaluate the change in representation in the union of selective voxels and the non-selective voxels in each hemisphere (Fig. 3c, f). That is, in each hemisphere and voxel subset we first computed the mean Euclidean distance across all categories from each child's MDS embedding of their first session to the embedding of their last session. We then compared the mean distances across the union of selective voxels to that in the non-selective voxels.

**Statistical analyses related to Fig. 4.** For the analysis in Fig. 4a, c, we used LMMs to test if distinctiveness in the union of selective voxels is related to behavior. The reported p-values are adjusted for multiple comparisons using the FDR procedure described for Fig. 1.

Figure 4a: While we acquired reading scores for both real words and pseudowords in our participants, we used the reading scores for pseudowords to test for a link between distinctiveness and behavior, because pseudowords were also used in the fMRI experiment. We used a random slopes model that can be specified as:

Pseudoword reading score (%) ~ distinctiveness for pseudowords + (distinctiveness for pseudowords | participant).

A random slopes model was used because (i) it fitted the data better compared to the random intercept model and (ii) the individual participant's slopes (random effects) visualize that there is a positive link between distinctiveness for pseudowords and reading score in most individuals.

Figure 4c: In this analysis we tested if there is a link between distinctiveness for adult faces in the union of selective voxels in lateral VTC and face recognition scores as assessed with the CFMT using adult faces. While we had also acquired data of a version of the CFMT using child faces (CFMT-child), we used data of the CFMT-adults to link it to distinctiveness for two reasons. First, performance on the CFMT-child was overall higher resulting in ceiling effects in some participants. Second, as the CFMT-adults is a widely used test, it enables comparison across studies. As such, for the data presented in Fig. 4c we used a random slopes model that can be specified as:

Face recognition score in CFMT-adults (%) ~ distinctiveness for adult faces + (distinctiveness for adult faces | participant).

While the random slopes model did not fit the data significantly better than the random intercept model, it enables visualizing the effect between face distinctiveness and face recognition in individual participants. There are no significant differences to the results of the present analysis when a random intercept model is used instead of the random slopes model.

**Predicting behavior from brain data.** We used a leave-one participant-out-cross validation (LOOCV) approach to test if we can predict behavior from brain data. That is, we computed a LMM that predicts behavior using distinctiveness on all sessions except for one participant that was left out in each iteration. Then, we used the LMM estimates to compute the predicted behavioral score for each session of the left-out participant. The prediction error was defined as the difference between the predicted and the actual behavioral score. We then repeated this procedure for all participants.

The box plots in Fig. 4b, d show the prediction error for three different models:

(i) Selective: LMM predicting behavior from distinctiveness over the union of selective voxels.
  B: Pseudoword reading score ~ distinctiveness for pseudowords across the union of selective voxels + (distinctiveness for pseudowords | participant);
  D: Face recognition score ~ distinctiveness for adult faces across the union of selective voxels + (distinctiveness for adult faces | participant);

(ii) Non-selective voxels: LMM predicting behavior from distinctiveness over the non-selective voxels.
  B: Pseudoword reading score ~ distinctiveness for pseudowords across the non-selective voxels + (distinctiveness for pseudowords | participant);
  D: Face recognition score ~ distinctiveness for adult faces across the non-selective voxels + (distinctiveness for adult faces | participant);

(iii) Selective & age: LMM predicting behavior from distinctiveness over the union of selective voxels with age as an additional factor.
  B: Pseudoword reading score ~ distinctiveness for pseudowords across the union of selective voxels + age + (distinctiveness for pseudowords | participant);
  D: Face recognition score ~ distinctiveness for adult faces across the union of selective voxels + age + (distinctiveness for adult faces | participant);

In the analyses in the scatterplots in Fig. 4b, d we calculated the difference in prediction error using a model that predicts behavior based on distinctiveness in the union of the selective vs. in the non-selective voxels. Next, we used one-sample t-tests to evaluate if the difference was significantly different from zero.

### Reporting summary

Further information on research design is available in the Nature Portfolio Reporting Summary linked to this article.

## Data availability

The data generated with this study have been deposited in Zenodo[66] https://doi.org/10.5281/zenodo.8366779. The processed data required to generate the main and supplemental figures are available at: https://github.com/VPNL/distributedVTCDevelopment and https://doi.org/10.5281/zenodo.8366779. This repository includes a directory excel-files with tables containing detailed statistics for the analyses shown in Supplementary Fig. 2-5. The raw data generated in this study are protected and not available due to data privacy laws. Raw data can be made available upon request from KGS or MN.

## Code availability

fMRI data were analyzed using the open source mrVista software package (available on GitHub: http://github.com/vistalab). Preprocessing of the functional data was performed using the code provided in: https://github.com/VPNL/fLoc. Synthetic T1 brain volumes were generated using the mrQ software package: https://github.com/mezera/mrQ. Original code to generate the main and supplementary figures and tables is available at: https://github.com/VPNL/distributedVTCDevelopment.

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

## Acknowledgements

We thank Laura Villalobos, Erica Yeawon Hwang, Savana Huskins, Alema Fitisemanu, and Philip Eykamp, Cat Davis and Tai Anthony McMillan for manually editing gray-white matter brain segmentations. We thank Caitlyn Estrada and Nancy Lopez-Alvarez for help with data collection, and Rachel Hinds for help with data entry and management. This project is funded by a fellowship of the German National Academic Foundation NO 1448/1-1 (MN); NIH grant RO1EY022318 (KGS); NIH training grant 5T32EY020485 (VSN); NSF Graduate Research Development Program DGE-114747 (JG); Ruth L. Kirschstein National Research Service Award F31EY027201 (JG). The funders had no role in study design, data collection and analysis, decision to publish, or preparation of the manuscript.

## Author contributions

M.N. collected data, developed, and coded the analysis pipeline, analyzed the data, and wrote the manuscript. V.S.N. and J.G. designed the experiment, collected data, and contributed to the manuscript. A.A.R., D.F. and H.K, collected the data, and contributed to the manuscript. K.G.S. designed the experiment, contributed to the analysis pipeline and data analyses, and wrote the manuscript.

## Funding

## Competing interests

The authors declare no competing interests.
