## [Peer Review File · Nature Communications]

Longitudinal development of category representations in ventral temporal cortex predicts word and face recognitionREVIEWER COMMENTS

Reviewer #1 (Remarks to the Author):

Nordt and colleagues test if distributed category representations across ventral temporal cortex (VTC) change with age and are linked to changes in perception in school-age children. They find increases and decreases in the distinctiveness of category representations with age, with as main results that representations for words become more distinct in the left hemisphere and for faces bilaterally, an effect carried by category selective voxels in VTC. Category distinctiveness in these voxels and regions predicts individual word and face recognition performance.

The multivariate approach offers an important missing perspective on the development of category selectivity across VTC, as nearly all work on this so far has focussed on subsets of category selective clusters even though it's been known for decades that category information is present throughout adult VTC so could contribute to development. A select number of studies have used a similar approach, but a unique element here is that this is longitudinal data, which offers improved detection power for both developmental changes in representation and brain behaviour correlations. Below I raise some concerns, most importantly about how the choice of ROIs shapes interpretations and how results relate to known developmental processes.

Defining category selective voxels I

> Debates in the literature about the role of distributed category representation often ask whether useful information about a specific category (e.g., faces) is present in voxels not maximally selective for this category, including voxels maximally selective for a completely different category (e.g., Haxby, Gobbini et al, Science, 2001). Here, the authors test a different question: namely whether category information is contained in cortex selective for ANY object category of 10 tested (termed 'unity of selective voxels') versus in cortex selective for no specific category at all (non-selective voxels). Whilst also interesting, this is a particular interpretation of category selectivity with some implications that require consideration:

First, it leaves unclear the degree to which representational change for a specific category (e.g., faces) within the unity of category selective voxels is driven by voxel clusters selective for that category (hypothesis 1, line 80) or is distributed across voxels selective for irrelevant categories (e.g., instruments, cars, etc.). Second, the odds stack against detecting change in category information in non-selective voxels (hypothesis 2, line 88), for one because this ROI will not just consist of well-responding voxels without any known category selectivity but also poor-responding voxels, likely taking up larger proportions as more categories are included in the design.

Given these considerations it would be interesting to see analyses that further pinpoint changes to *category-relevant* category selective voxels versus remaining irrelevant or no category selective cortex in VTC; This would allow for a different and more sensitive way to discriminate between hypotheses 1 and 2, and be relevant to previous research making similar distinctions. There are many possible approaches. One would be to run the analyses in Fig 2,3&4 not just in the union of category selective voxels and outside of it, but also separately for each category (i.e., test the degree to which face-selective voxels vs. non-face selective voxels contribute to changes in the distinctiveness of face representations as well as of other categories).

Defining category selective voxels II

> Why is the medial VTC not included in the analysis and would this change the results?

Defining category selective voxels III

> Category selective voxels seem to be defined per time point and participant. If correct, different voxel populations are probably included in category selective vs non-selective ROIs at different timepoints. This in itself could drive differences in pattern responses within these ROIs irrespective of any change in the underlying cortical representation. Do the effects change if identical ROIs are used across timepoints (say, those based on the latest measure)?

Novelty in relation to established findings of developmental change in VTC

> An important result is that increased distinctiveness of word and face representation across category selective voxels in specific regions can be used to predict improved word and face recognition. Many previous studies by this group and others have likewise related a change in extent and selectivity of face and word-selective clusters in these regions to face and word recognition performance. This begs the question of how current results reflect this known process quantified in a new way, or a new independent process. To address this, could the authors for example compare how developmental changes within word/face selective regions vs wider (category selective or non-selective) cortex in VTC explain unique aspects of performance?

Minor comments

> If the voxel time course correlation that can be obtained at earlier timepoints is inherently lower due to noisier data, this reduces power to detect category distinctiveness for other reasons than development alone. I appreciate that signal to noise is considered in the LMM, but a different correction might be more relevant for the MVP analysis. For example, would it be possible to scale measures by a noise ceiling, i.e., the theoretically highest time course correlation possible in the participant at time A? This might be computed from correlations between identical stimuli repeated across runs, potentially for a stationary category such as cars, and be either taken as covariate or used to convert correlation plots to proportion of maximum correlation?

> Please scatter the model-predicted versus obtained behavioural scores as individual datapoints in Fig 4B&D. It would also be nicer to plot bar plots across the manuscript (e.g., Fig 1d, 2) in a manner that informs about underlying datapoints.

> Missing bracket line 468.

> The finding of a change in house representation in the IVTC is described as "unexpected". I do not see why, as this is in line with a distributed representation of category information across the VTC outside category selective clusters, and I am not aware of a reason to expect this to be restricted within the lateral VTC?

> the category selective voxels are identified based on an arbitrary threshold of $t\text{-value} > 3$ for the contrast [category - all other categories except categories of the same domain]. What motivated this threshold, is the pattern of results robust across different thresholds?

Reviewer #2 (Remarks to the Author):

The present manuscript reports a functional MRI study addressing the developmental changes in the categorical object representation in the visual ventral temporal cortex (VTC), between 5 and 13 years of age. Focusing on the lateral aspect of the VTC, the study measures the distinctiveness between categories: how much the neural response pattern for a category differs from the neural response patterns for other categories. Results show that 1) distinctiveness changes (increases or decreases) over time for some categories but not for others, 2) changes are driven by voxels in category-selective clusters of the lateral VTC, and 3) distinctiveness for faces and words in the lateral VTC correlates

with, and predicts, behavioral performance in face recognition and pseudoword respectively.

- Does the work support the conclusions and claims, or is additional evidence needed?
- Are there any flaws in the data analysis, interpretation and conclusions? - Do these prohibit publication or require revision?
- Is the methodology sound? Does the work meet the expected standards in your field?
- Is there enough detail provided in the methods for the work to be reproduced

The manuscript is well written; it tackles an important question with sound methodology and analyses. The results are statistically reliable and interesting. Methods and analyses are described with enough detail for replication. However, there are some important issues that need to be considered before the implications of the results and the general impact of the work for understanding development can be evaluated. First, a big part of the data (the medial VTC) and of the categories (inanimate objects) are not considered, and it is not clear why. The analysis in the medial VTC could clarify the effects for categories that, in the current analyses, yield inconclusive null results. Second, the results support the strong claim that developmental changes in visual categorical object representation are driven by changes in selective voxels, as opposed to more distributed cortical representations. This seems contrary to what has been suggested by other results in infants (Spriet et al., 2022; 119.8: e2105866119). In addressing a possible conflict, the authors may need to recognize that the developmental change described here may be less general than it may appear, possibly limited to a certain temporal interval (or developmental stage: late childhood to adolescence). Third, there are a number of effects that are not so straightforward and are not explained (e.g., effect of age in the relationship between face distinctiveness and recognition; lateralization of changes in the number representation), and methodological choices that need to be better justified (e.g., definition of selective voxels, variables in the correlation). Below, I discuss these and a few other issues, hoping my comments will help the authors improve the manuscript.

First, it is not clear why the study focuses on the lateral VTC, leaving aside the medial VTC. The lateral VTC hosts selective regions for face, limbs and words, but selectivity for other categories (notably, objects and places) can rather be found in the medial VTC (see e.g., Grill-Spector & Weiner, 2014). For categories such as cars, corridors and instruments, the study reports null effects (Fig. 1-2), which are inconclusive –and for no reason, given that data are available. Do those null effects reflect genuine category-specific differences in terms of developmental changes (neural representations for certain categories, but not others, continues to evolve in late childhood/adolescence)? Or do they simply reflect poor measurement in regions that might show effects for those categories of inanimate objects? Developmental changes of face and word representation, which is the main goal of the study, can be better understood by comparison with other categories.

One intriguing result of the study is the lack of evidence for developmental changes (in distinctiveness) in non-selective voxels. Is it possible that the time window in which the effects have been tested (late childhood to adolescence) is not a sensitive period to observe such changes? Recent studies suggest a quite different story. Spriet et al. (2022 PNAS) tested the developmental changes in visual object categorization using eight different categories, in infants from 4 months to 18 months. One of the findings is that, as infants grow older, they show (behaviorally) to represent more and more categories and this change is correlated with categorical object representation in ever larger aspects of the ventral cortex. Thus, this study suggests that the development of object category representation proceeds with an increase in the distributedness of the representation. Of course, children/adolescents and infants are different “species” but that’s the question. In concluding for one hypothesis (e.g., change in category distinctiveness is driven by selective voxels), it should be considered that the current time window might be sensitive to only some effects, while other effects (e.g., changes in more distributed representations) may occur at a different time. What kind of development is the development measured in late childhood/adolescence? How does it relate to developmental phenomena occurring earlier (in early infancy)? This speaks to the generality of the current results with respect to explaining the mechanisms of longitudinal development of visual

category representation.

To test the link between cortical development and behavioral development concerning face and word processing, the authors carried out correlational analysis considering performance in face recognition/word reading and the distinctiveness of those categories. However, if the goal is to test the link between the two developments (in cortical representation and in behavioral performance), why not consider the changes in category distinctiveness (to be clear, the measure plotted for example in Fig. 1d), rather than the distinctiveness itself. This analysis would support the authors' conclusion that "developmental improvements in reading and face recognition abilities are linked to increases in word and face distinctiveness" (lines 426-427).

The relationship between word distinctiveness and reading was independent from the effect of age, but the relationship between face distinctiveness and face recognition was dependent on age (the effect was no longer significant in the left hemisphere and there was only a trend in the right hemisphere). On this analysis, the authors conclude that distinctiveness of face representations in lateral VTC predicts face recognition performance. How does this conclusion account for the fact that the effect is not significant when age is considered?

For the second analysis, the authors separate selective and non-selective voxels. First, why did the authors select as a criterion, a voxelwise effect of $t > 3$? Is this a common/standard criterion for voxelwise selectivity? And how did the authors treat those voxels that were selective ($t > 3$) for more than a category? And which of the 10 categories had selective voxels in the lateral VTC? How was the distribution of those voxels? Do they give rise to the classic category-selective regions that one would find with a functional localizer? A good sanity check would be to show category-selective clusters on the brain...

From Fig. 1 and 2, changes for numbers are stronger in the right hemisphere. However, the MDS embeddings with unions of selective voxels (Fig. 3) show that representations of pseudowords and numbers increases from age 5 to age 17, particularly in the left hemisphere. Isn't there a conflict?

Limbs: distinctiveness decreases over time when considering all the voxels (Fig. 1) but then this effect is significant neither in selective or in non-selective voxels. What does this mean? Could it be the case that distinctiveness for this category is contributed by both selective and non-selective voxels?

From Fig. 2 it looks like distinctiveness for child faces increases more in left than in the right hemisphere. Is this statistically true? If so, does this result matter?

Missing words in line 402?

Reviewer #3 (Remarks to the Author):

This is a fascinating dataset and an equally intriguing set of hypotheses. The links made between changes in distinctiveness and recognition behaviour is a welcome one and bolsters the claims made. I am enthusiastic about the data, but have a number of queries that I feel would only strengthen an already strong manuscript, which I outline below (in no particular order).

1. Can the authors confirm that analyses were performed on voxels and not surfaces nodes (or indices if you prefer)? Fig 1.b shows a surface representation of left lateral VTC. If this is for illustrative purposes only, can it be stated thus?
2. Unless I missed it, were off diagonal values averaged together in the calculation of distinctiveness (Fig 1.a)?
3. Indicators of significance would be useful in Fig. 1d
4. The lack of hemispheric difference in the distinctiveness of both child and adult faces was a surprise

to me. The left hemisphere preference for pseudowords is clearly present, but I would have expected a commensurate change in the right hemisphere. The authors do not discuss this and its implications for the debate about lateralisation of function and I believe this is a missed opportunity.

5. Paragraph starting lines 178: The authors mention that motion was added as a predictor into the LMM, but weren't the values that go into the LMM already computed on data that had been motion corrected (see methods)? This needs to be clarified.

6. The analysis of selective V non-selective requires deeper investigation and broader discussion:

a. Why was a t-value of 3 selected?

b. What p-level does this reflect?

c. If changes in distinctiveness are truly only present within category-selective voxels, does the degree of distinctiveness increase as selectivity increased (e.g., if a voxel is selective at $t=6$ is category representation more distinctive than a voxel at $t=3$). Again, this feels like a missed opportunity to fully explore these effects and provide a comprehensive picture.

d. At what levels of selectivity does distinctiveness break down? Would a voxel at $t=2.99$ really not show distinctiveness?

7. Related to point 4 above, the analysis of distinctiveness in category-selective voxels (Fig. 2) shows higher levels of average distinctiveness in the left hemisphere over the right for both adult and child faces – this needs to be at least highlighted and discussed.

8. Again, I remain enthusiastic about the manuscript, but too much is made of the difference between the prediction error of selective V non-selective voxels ($p=0.045$).

9. Paragraph beginning line 397:

a. There is an incomplete sentence on line 400-401 that needs rectifying.

b. There needs to be a discussion of the fact that when age is added for to the LMM this result is no longer significant at $p < .05$. This is important because the authors ultimately put forward the case for distinctiveness being restricted to selective voxels, but that finding appears only to be consistent for pseudowords.

c. There are no corrections for multiple comparisons applied to the results reported.

Point-by-point response

Point-by-Point Response to the Reviewers

We thank the reviewers for their comments, which we reply to in full below. Our responses to the reviewers' comments are indicated in blue under each comment. We have highlighted changes in the manuscript using the track changes function.

REVIEWER COMMENTS

Reviewer #1 (Remarks to the Author):

Nordt and colleagues test if distributed category representations across ventral temporal cortex (VTC) change with age and are linked to changes in perception in school-age children. They find increases and decreases in the distinctiveness of category representations with age, with as main results that representations for words become more distinct in the left hemisphere and for faces bilaterally, an effect carried by category selective voxels in VTC. Category distinctiveness in these voxels and regions predicts individual word and face recognition performance.

The multivariate approach offers an important missing perspective on the development of category selectivity across VTC, as nearly all work on this so far has focussed on subsets of category selective clusters even though it's been known for decades that category information is present throughout adult VTC so could contribute to development. A select number of studies have used a similar approach, but a unique element here is that this is longitudinal data, which offers improved detection power for both developmental changes in representation and brain behaviour correlations. Below I raise some concerns, most importantly about how the choice of ROIs shapes interpretations and how results relate to known developmental processes.

We thank the reviewer for their positive view on our research.

Defining category selective voxels I

1. Debates in the literature about the role of distributed category representation often ask whether useful information about a specific category (e.g., faces) is present in voxels not maximally selective for this category, including voxels maximally selective for a completely different category (e.g., Haxby, Gobbini et al, Science, 2001). Here, the authors test a different question: namely whether category information is contained in cortex selective for ANY object category of 10 tested (termed 'unity of selective voxels') versus in cortex selective for no specific category at all (non-selective voxels). Whilst also interesting, this is a particular interpretation of category selectivity with some implications that require consideration:

First, it leaves unclear the degree to which representational change for a specific category (e.g., faces) within the unity of category selective voxels is driven by voxel clusters selective for that category (hypothesis 1, line 80) or is distributed across voxels selective for irrelevant categories (e.g., instruments, cars, etc.). Second, the odds stack against detecting change in category information in non-selective voxels (hypothesis 2, line 88), for one because this ROI will not just consist of well-responding voxels without any known category selectivity but also poor-responding voxels, likely taking up larger proportions as more categories are included in the design.

Given these considerations it would be interesting to see analyses that further pinpoint changes to **category-relevant** category selective voxels versus remaining irrelevant or no category selective cortex in VTC; This would allow for a different and more sensitive way to discriminate between hypotheses 1 and 2, and be relevant to previous research making similar distinctions. There are many possible approaches. One would be to run the analyses in Fig 2,3&4 not just in the union of category selective voxels and outside of it, but also separately for each category (i.e., test the degree to which face-selective voxels vs. non-face selective voxels contribute to changes in the distinctiveness of face representations as well as of other categories).

We agree with the reviewer that it would be informative to examine the changes in distinctiveness in voxels selective to one particular category (for instance, faces), in addition to the approach chosen in the manuscript (the union of selective voxels). Therefore, we have run an additional analysis to address this suggestion: In this analysis, we created disk-ROIs (radius=10mm) for each category-selective ROI (for instance the word- or face-selective region) for each subject. To define the disk ROIs, we used all ROIs of that contrast in a given subject (i.e. word-selective ROI in session 1, 2 and 3 of subject A), which had individually been defined in the respective subject's functional sessions. Then, we computed the average center coordinates of the ROIs of all sessions belonging to that subject. We next created the disk ROI centered on those mean coordinates. This approach was chosen as (i) it ensures that the voxels will show selectivity to one particular category, (ii) it ensures that the number of included voxels is constant across sessions of a subject, and (iii) provides a definition of the ROI that is unbiased to any session and is independent from the conducted analyses. We show the results for the word-selective ROIs pOTS-words and mOTS-words, the face-selective ROIs pFus-faces and mFus-faces, the limb-selective ROI OTS-limbs and the place-selective ROI CoS-places. For the word-selective ROIs we restrict our analyses to the left hemispheres, as few subjects had the ROIs in the right hemisphere.

The results of this analysis can be found below. In addition, we have included these results as new Fig. S5 and have added a description of the analyses in the methods section. One finding that is visible when examining these new plots is that in most ROIs we see development of distinctiveness for the category that was used to define the ROI. That is, for words-selective ROIs (pOTS-words and mOTS-words) we see an increase in distinctiveness for words. For face-selective ROI (pFus-faces and mFus-faces) we see an increase in distinctiveness for faces and for the place-selective ROI (CoS-places) we see an increase in distinctiveness for houses. In case of the limb-selective ROI the decrease in distinctiveness for limbs is non-significant. In addition to the development for the ROI-defining category, in some cases other categories show development as well. For instance, in the word-selective ROI pOTS-words we also find a significant increase in distinctiveness for numbers and in the more anterior word-selective ROI mOTS-words we find an increase in distinctiveness for faces. The posterior face-selective ROI shows an increase in distinctiveness for words, and the anterior face-selective ROI shows an increase in distinctiveness for cars. The limb-selective ROI shows an increase for faces and places, and the place-selective ROI shows an increase in distinctiveness for limbs. In sum, these analyses suggest that distinctiveness develops for categories that were used to define the ROI, but that development is not limited to those.

We refer to the results of these analyses in the text as follows:

"To further investigate how development within clustered category-selective regions contributes to development of distinctiveness, we also conducted an analysis where we created independent disk-ROIs for each category-selective ROI (e.g., word-selective region) for each subject. We chose this approach to ensure that: (i) we examine information across voxels selective to one category, (ii) we use a constant number of included voxels across sessions of a participant and (iii) that the voxel selection will not be

biased to a particular session. Results show that distinctiveness develops for the category that was used to define the ROI (e.g., there is an increase for distinctiveness for words in the word-selective ROI), but also that development is not limited to the preferred category (Fig S5).”

Fig. S5 Development of distinctiveness in category-selective disk ROIs.

Changes in distinctiveness in disk ROIs centered on (A) pOTS-word (lh: n=28 subjects, 125 sessions), (B) mOTS-word (lh: n=23 subjects, 103 sessions), (C) OTS-limbs (lh: n=28 subjects, n=126 sessions; rh: n=28 subjects, n=126 sessions). (D) pFus-faces (lh: n=27 subjects, 120 sessions; rh: n=22 subjects, 98 sessions), (E) mFus-faces (lh: n=23 subjects, 107 sessions; rh: n=24 subjects, 106 sessions), (F) CoS-places (lh: n=29 subjects, 128 sessions; rh: n=29 subjects, 128 sessions). For pOTS-words and mOTS-words data is shown only for the left hemisphere due to low n in the right hemisphere. LMM slopes indicating change in distinctiveness per year in category-selective disk ROIs (LMM relating distinctiveness to age, with tSNR as an independent factor, and participant as random effect) for each category. Error bars: 95% confidence interval (CI). If the CI does not cross the y=0 line, the change in distinctiveness is significantly different than 0. Disk-ROIs (radius=10mm) were created for each category-selective ROI for each subject. To define the disk ROIs, ROIs of all sessions of that contrast in a given subject (i.e., word-selective ROI in session 1, 2 and 3) were used, which had individually been defined in the respective subject's functional sessions. Then, the average center coordinates of the ROIs of all sessions belonging to that subject was computed and the disk ROI was centered on those mean coordinates. This approach was chosen as (i) it ensures that the voxels will show selectivity to a certain category, (ii) it ensures that the number of included voxels is constant across sessions of a subject, (iii) provides an independent definition of the ROI, and (iii) minimizes bias towards a particular session.

We would also like to respond to the concern raised by the reviewer that the lack of development in the non-selective voxels may be driven by poor-responding voxels, rather than well-responding voxels without any known category selectivity. We agree that this is an important aspect and we have added a new control analysis (new Fig. S2, see below) to address this concern. In this new analysis, we have examined the change in category distinctiveness in subsets of the union of the selective voxels and of the non-selective voxels that are matched on the amount of variance explained by the GLM as we reasoned that noisy voxels will have less variance explained. 300 voxels were chosen for each voxel subset (i) to match the number of voxels across subsets and (ii) to ensure the analysis can be conducted in all sessions, including sessions with a low number of selective voxels. We have added a description of this new analysis in the Methods.

This analysis shows that in these two subsets of voxels (union of selective/non-selective) that do not differ in variance explained, we find largely the same results as in our main analysis presented in Fig. 2: All significant developmental effects in the union of selective voxels are also present here (maroon in Fig S2). In case of the non-selective voxels, we replicate no significant development (gray in Fig S2), except that in this subset there is a significant development of distinctiveness for numbers. As such, these results suggest that the lack of a development in the non-selective voxels observed in Fig. 2a is not driven by poor-responding voxels.

Fig S2. Development of distinctiveness in subsets of the union of selective voxels and the non-selective voxels in lateral VTC that are matched for variance explained.

Bars indicate the change in category distinctiveness per year (LMM relating distinctiveness to age and tSNR, with participant as a random effect, $n=128$ sessions, 29 children) in different subsets of voxels of lateral VTC that are matched for variance explained. *Maroon bars*: a subset of the size of 300 voxels out of the union of voxels in lateral VTC defined in each session. *Gray bars*: a subset of 300 voxels out of the non-selective voxels of lateral VTC defined in each session.

Category-selectivity was computed by contrasting responses to a category vs. all other categories except the other category from the same domain (e.g., numbers vs. all other categories except words). A voxel was defined as selective to a category when $t > 3$. Darker colors: left hemisphere. Lighter colors: right hemisphere. Error bars: 95% CI. If the CI does not cross the $y=0$ line, the change in distinctiveness is significantly different than 0.

We have included these results in the main text as follows:

“Additionally, we repeated the analyses in subsets of the union of the selective and the non-selective voxels that were matched for variance explained and for the number of included voxels. This analysis showed the same pattern of significant development of distinctiveness for the union of the selective voxels, and no evidence for significant development for the non-selective voxels, except for a significant increase in distinctiveness for numbers (Fig. S2) suggesting that the lack of development in the non-selective voxels is not driven by poorly responding voxels.”

Defining category selective voxels II

2. Why is the medial VTC not included in the analysis and would this change the results?

Based on this comment and similar suggestions made by the other reviewers we have added new analyses examining the longitudinal development of distributed representations in medial VTC to the manuscript. These additional analyses can be found in Fig. 1e, Fig. 2b and Supplemental Figures S7 & S8.

In the introductory sentences of the results section, we now write:

“To assess the nature of distributed category representations in children, we computed the distributed pattern of responses for each category. As face-, limb-, and word-selective regions are located in lateral aspect of VTC^{7,8,11,30}, we divided VTC into its lateral and medial partitions and because development of word- and face-selectivity varies across hemispheres^{16,18,21,31}, we measured in each child and session distributed responses to each of the 10 categories, separately for lateral and medial VTC in each hemisphere.”

We have added a new panel to Fig. 1 showing the development of distinctiveness in medial VTC:

Fig. 1e: The development of distinctiveness by age in all voxels in medial VTC. Y-axis: LMM slopes for each category; LMM relates distinctiveness to age, with tSNR as an independent factor, and participant as random effect (n=128 sessions, 29 children). Asterisks indicate significant development (p<0.05). Circles around asterisks indicate significant development after FDR-correction.

We have added the following paragraph in the results section describing this new panel of Fig. 1:

“We next examined development of distinctiveness in medial VTC. Consistent with the first hypothesis, we find increases in distinctiveness for houses in left and right medial VTC (left: $\beta_{age}=0.023$, $t(125)=2.76$, $p_{FDR}=0.034$, right: $\beta_{age}=0.03$, $t(125)=3.61$, $p_{FDR}=0.009$). Consistent with the second hypothesis, we also find development for a category without a clustered region in medial VTC: Distinctiveness for adult and child faces increased significantly in the left hemisphere in medial VTC (adult faces: $\beta_{age}=0.019$, $t(125)=3.05$, $p_{FDR}=0.028$, child faces: $\beta_{age}=0.019$, $t(125)=2.76$, $p_{FDR}=0.034$). We find no evidence for other significant development in medial VTC (Table 2).”

In addition, we have added this new panel to Fig. 2. It shows the development of distinctiveness in different voxel subsets (union of the selective, non-selective voxels) in medial VTC.

Fig. 2b: This analysis examines changes in distinctiveness in different subsets of voxels in medial VTC: the union of the selective voxels and the remainder, the non-selective voxels. In the union of the selective voxels, this analysis reveals marginally significant increases in distinctiveness for houses in the right hemisphere. In the non-selective voxels, the analysis reveals a marginally significant increase in distinctiveness for adult and child faces in the left hemisphere and for numbers in the right hemisphere. These effects don't survive the FDR-correction for multiple comparisons.

We report the results of these analyses in the text as follows:

"We next examined the development of distinctiveness in these subsets of voxels in medial VTC. As in lateral VTC, the union of the selective voxels comprised less voxels compared to the non-selective voxels (see legend Fig. 2b). Examining changes in distinctiveness in the union of the selective voxels revealed an increase in distinctiveness for houses in right medial VTC ($\beta_{age}=0.028$, $t(125)=3.03$, $p_{FDR}=0.12$), which was not significant after FDR-correction (Fig. 2b-maroon & pink bars). In the non-selective voxels (Fig. 2b-gray bars), there were increases in distinctiveness for faces in the left (adult faces: $\beta_{age}=0.012$, $t(125)=2.41$, $p_{FDR}=0.35$; child faces: $\beta_{age}=0.013$, $t(125)=2.06$, $p_{FDR}=0.41$) and for numbers in the right hemisphere ($\beta_{age}=0.011$, $t(125)=2.16$, $p_{FDR}=0.41$), which were not significant after FDR-correction. We found no evidence for other significant effects (Table 5,6). Result were largely the same for a range of thresholds to define selective voxels (Fig. S4)."

Defining category selective voxels III

3. Category selective voxels seem to be defined per time point and participant. If correct, different voxel populations are probably included in category selective vs non-selective ROIs at different timepoints. This in itself could drive differences in pattern responses within these ROIs irrespective of any change in the underlying cortical representation. Do the effects change if identical ROIs are used across timepoints (say, those based on the latest measure)?

The reviewer raises an important point. In the current analysis reported in the manuscript, the voxel sets (the union of category-selective voxels and that of non-selective voxels) are defined per time point in each participant. The motivation for defining selective voxels in each time point was to minimize bias, which can be introduced when one session is used to define selective voxels, as in that case, the session in which voxels are defined is treated differently compared to all other sessions.

That being said, we agree with the reviewer that it is also informative to examine changes in distinctiveness over sets of voxels that are constant across sessions of a given subject. The analyses presented in Fig. 1d,e are examples for using the same voxels across sessions, as in our analyses each functional session of a given participant is aligned to its own-n-year-anatomical average, therefore ensuring that the ROI is constant across sessions.

To address the mentioned concern also with regard to voxel subsets defined by category-selectivity, we have conducted the disk-ROI analysis, which we have presented above (see our response to comment 1, new Fig. S5). As described above, this analysis (i) uses constant voxels across sessions, (ii) provides an independent definition of the ROIs, and (iii) minimizes bias because all sessions are treated the same way. Across ROIs, this analysis shows development for the same categories as in the analysis presented in the main figures: We find an increase in distinctiveness for words and numbers (in the word-selective ROIs), an increase in distinctiveness for faces (in the face-, limb- and word-selective ROIs), an increase in distinctiveness for houses (in the place- and limb-selective ROIs). The only effect found in the main analysis, that is not found here, is the decrease in distinctiveness for limbs. We speculate that this effect may be driven by both the selective and the non-selective voxels (see also our response to reviewer 2's comment 7). In addition, these analyses also reveal an increase in distinctiveness for limbs in the place-selective ROI and an increase for distinctiveness for cars in the face-selective ROI mFus-faces.

Novelty in relation to established findings of developmental change in VTC

4. An important result is that increased distinctiveness of word and face representation across category selective voxels in specific regions can be used to predict improved word and face recognition. Many previous studies by this group and others have likewise related a change in extent and selectivity of face and word-selective clusters in these regions to face and word recognition performance. This begs the question of how current results reflect this known process quantified in a new way, or a new independent process. To address this, could the authors for example compare how developmental changes within word/face selective regions vs wider (category selective or non-selective) cortex in VTC explain unique aspects of performance?

We agree with the reviewer that it is relevant to compare how the link between category distinctiveness and behavior compares to links between category selectivity and behavior. To address this question, we have repeated the analyses shown in the scatterplots in Fig. 4a,c but used the number of selective voxels instead of category distinctiveness as predictor of behavior. That is, in case of words and reading we used the number of word-selective voxels in left lateral VTC to predict behavior. In case of faces and face recognition we used the number of (adult) face-selective voxels in right lateral VTC to predict adult face recognition behavior. Category-selective voxels were defined as voxels with a t-value > 3 for the contrasts: (i) words vs all other categories except numbers, and (ii) adult faces vs all other categories except child faces. In addition, we also tested if the size of individually defined functional regions of interest (pOTS-words and pFus-faces) were related to behavior. The results of this analysis are shown below and have been added as Supplemental Fig. S10.

Fig S10. No significant link between behavior and the number of word- and face-selective voxels or the size of word- or face-selective ROIs after correcting for age. (A) Linear mixed model (LMM) with random slopes and intercepts relating reading performance of pseudowords (Woodcock Reading Mastery Test, WRMT) to the number of voxels selective to pseudowords in anatomically defined left lateral VTC ROIs. Selective voxels are defined by a t -value > 3 (contrast: pseudowords vs. all other categories except numbers). (B) Same as (A) but for volume of the left word-selective pOTS-words. (C) LMM with random slopes and intercepts relating performance in the Cambridge Face Memory Test (CFMT) to the number of voxels selective to adult faces in anatomically defined right lateral VTC ROIs. Selective voxels are defined by a t -value > 3 (contrast: adult faces vs. all other categories except child faces). The LMM shows a significant effect of the number of selective voxels (see parameters indicated in the figure). This effect is no longer significant once age is added to the model (LMM with factors age and number of selective voxels: $\beta_{nr \text{ of Voxels}}=0.0065$, $t(79)=1.16$, $p=0.25$; $\beta_{age}=3.78$, $t(79)=7.48$, $p<0.001$). (D) Same as (C) but for the number of voxels in the face-selective ROI pFus-faces in the right hemisphere. The LMM shows a significant effect of the number of selective voxels (see parameters indicated in the figure). This effect is no longer significant once age is added to the model (LMM with factors age and number of selective voxels: $\beta_{nr \text{ of Voxels}}=0.01$, $t(56)=1.22$, $p=0.23$; $\beta_{age}=3.81$, $t(56)=5.94$, $p<0.001$).

Model parameters indicated in the bottom. Each dot is a session; Dots are colored by participant. Colored lines: individual slopes and intercepts. Thick gray line: LMM prediction non including age in the model. Shaded gray: 95% CI.

These results reveal that reading performance is not significantly linked to either the number of word-selective voxels in left lateral VTC or to the size of the left word-selective pOTS-words (Fig S10a,b). While there is a significant link between face recognition performance and the number of (adult) face-selective voxels in right lateral VTC and face recognition, as well as the size of right pFus-faces these relationships are no longer significant once age is added to the model. As such, these results suggest that distinctiveness for words and faces is a more parsimonious predictor for reading and face recognition, respectively, compared to the number of selective voxels for the given category.

We have included a brief description of these findings in the manuscript. In the section describing the link between distinctiveness for words and reading performance we added the following sentences:

“We also tested if a similar link exists between (i) the number of word-selective voxels in left lateral VTC or (ii) the size of the word-selective region pOTS-words in the left hemisphere and reading performance for pseudowords (Fig S10AB). However, reading performance was neither significantly related to the number of selective voxels for pseudowords, nor the size of the left word-selective pOTS-words, suggesting that word distinctiveness in left lateral VTC is a better predictor of reading performance than the other metrics.”

Similarly, in the section describing the link between distinctiveness for adult faces and face recognition performance we added the following sentences:

“As prior research has also suggested a link between face recognition performance and number of face-selective voxels²¹ we also examined this relation in our data. We found that face recognition was also significantly linked to the number of face-selective voxels and the size of the right face-selective pFus-faces (Fig S10C,D), but these relationships were no longer significant once age was added to the model.”

Minor comments

5. If the voxel time course correlation that can be obtained at earlier timepoints is inherently lower due to noisier data, this reduces power to detect category distinctiveness for other reasons than development alone. I appreciate that signal to noise is considered in the LMM, but a different correction might be more relevant for the MVP analysis. For example, would it be possible to scale measures by a noise ceiling, i.e., the theoretically highest time course correlation possible in the participant at time A? This might be computed from correlations between identical stimuli repeated across runs, potentially for a stationary category such as cars, and be either taken as covariate or used to convert correlation plots to proportion of maximum correlation?

We agree with the reviewer that it is important to make sure that the observed developmental effects are not due to underlying noise. Unfortunately, the proposed idea of computing correlations between identical stimuli across runs is not possible with the current dataset as in our experiment each image stimulus was only presented once (there are no repeated images). However, we are confident that the observed effects are not due to noisier data at earlier timepoints due to the following reasons: 1.) The main analysis presented in Fig. 1 shows a decrease in distinctiveness for limbs with age. That is, distinctiveness for limbs was lower in older children compared to younger children. As such, this result contradicts the idea that distinctiveness is lower in earlier timepoints due to noise. 2.) As the reviewer noticed, we have tested for effects of both tSNR and motion during scanning in our analyses. Adding motion as a predictor to the LMM did not significantly contribute to the model fit except for

distinctiveness for string instruments (no significant contribution of age to distinctiveness of string instruments with or without adding motion). Adding tSNR as a predictor to the LMM contributed to the model independent from age for several categories. Thus, we report age-related changes in distinctiveness that are independent from tSNR. 3.) Additionally, we added a new control analysis presented in our response to comment 1, where we controlled for variance explained between selective and non-selective voxels. This analysis replicates our findings showing development of distinctiveness in the selective voxels and no significant development in the non-selective voxels. These results indicate that the lack of development in the non-selective voxels is unlikely to be driven by lower variance explained (higher noise). Therefore, we are confident that the presented developmental effects are not driven by noise.

6. Please scatter the model-predicted versus obtained behavioural scores as individual datapoints in Fig 4B&D. It would also be nicer to plot bar plots across the manuscript (e.g., Fig 1d, 2) in a manner that informs about underlying datapoints.

We thank the reviewer for this suggestion and have replaced Figures 4b,d with boxplots showing the distribution of the data.

The bar plots in Figure 1d,e and 2 represent the slope of the linear mixed model (LMM, and confidence interval) that indicates the change in distinctiveness per year. As such there is one slope per category. In these analyses a random intercept (and fixed slopes) model fit the data better than random slopes models. Therefore, only one slope is obtained to characterize the change in distinctiveness for each category.

7. Missing bracket line 468.

We have added the missing bracket.

8. The finding of a change in house representation in the lVTC is described as "unexpected". I do not see why, as this is in line with a distributed representation of category information across the VTC outside category selective clusters, and I am not aware of a reason to expect this to be restricted within the lateral VTC?

This is a fair point, we have removed the word "unexpected" and rephrased the sentence, which now reads: *"Given that place-selective regions are located in the medial part of VTC^{10,12}, finding developmental increases in distinctiveness for houses in lateral VTC shows that increases in distinctiveness can also occur in parts of VTC that don't have a clustered region for that category."*

9. the category selective voxels are identified based on an arbitrary threshold of t-value>3 for the contrast [category - all other categories except categories of the same domain]. What motivated this threshold, is the pattern of results robust across different thresholds?

We thank the reviewer for raising this point. We agree that it is important to explain the choice of the threshold. A threshold of a t-value > 3 was used due to the following reasons. First, prior research has shown that category-selective regions can be defined reliably in individual subjects including both children and adults using this threshold (Finzi et al., 2021; Gomez et al., 2017, 2018; Natu et al., 2016; Nordt et al., 2021). In addition, a t-value of 3 ensures that the effect is 3 standard deviations from the mean. We have added a justification of the threshold to the methods section of the manuscript.

To ensure that the observed effects are robust across different thresholds, we have repeated the analyses in Fig. 2 for varying t-thresholds. That is, we ran the same analyses but used 5 different t-values t>1, t>2, t>3 (as in the prior analyses) t>4 and t>5 at threshold to define the union of the selective

voxels. We have added the results of these analysis as new Supplemental Fig. S3 for lateral VTC and Supplemental Fig. S4 for medial VTC.

Lateral VTC

Lateral VTC

New Fig. S3: Black squares (lh) and gray diamonds (rh) indicate the change in category distinctiveness per year (LMM relating distinctiveness to age and tSNR, with participant as a random effect, $n=128$ sessions, 29 children) in the union of voxels in lateral VTC. The x-axis shows the t-value that was used as a threshold to define a voxel as category selective. The results shown for at-value ≥ 3 (highlighted in yellow) correspond to the main analysis presented in Fig. 2 of the manuscript.

This analysis reveals that the developmental effects reported in Fig. 2 are largely stable across different thresholds used to define the union of selective voxels in lateral VTC. That is, for most categories we see (i) the same sign of change in distinctiveness (consistently positive or consistently negative) across different t-thresholds, and (ii) that the significance of effects is largely maintained across t-thresholds even if the numeric values of the change in distinctiveness are not identical.

However, we also notice a few instances where changes in the threshold influence the results. As such, we see that for numbers in the left hemisphere the increase in distinctiveness with age is significant for high t-values ($t > 4$ and $t > 5$). In case of distinctiveness for limbs we find that in the right hemisphere, the decrease in limbs is significant for low t-values ($t > 1$). This effect is in line with the significant decrease of distinctiveness for limbs in the right hemisphere in all voxels in lateral VTC (Fig 1). In case of faces, the analysis shows that the developmental increase of distinctiveness for faces in the right hemisphere is not significant for high t-values ($t > 4$ or $t > 5$). Finally, the change in distinctiveness for houses in the right hemisphere is no longer significant in the right hemisphere when defining selective voxels by $t > 5$.

We now link to these new analyses in the results section:

“We tested if these results change if the union of the selective voxels and the non-selective voxels are defined using different thresholds (Methods). We found that the results are largely robust across a range of t-thresholds defining selectivity (Fig. S3).”

Medial VTC

Medial VTC

New Fig. S4: Black squares (lh) and gray diamonds (rh) indicate the change in category distinctiveness per year (LMM relating distinctiveness to age and tSNR, with participant as a random effect, $n=128$ sessions, 29 children) in the union of voxels in medial VTC. The x-axis shows the t-value that was used as a threshold to define a voxel as category selective. The results shown for a t-value ≥ 3 (highlighted in yellow) correspond to the main analysis presented in Fig. 2 of the manuscript.

In medial VTC we also find that the effects are largely stable across different t-thresholds. Again, the results show that for most categories the same sign of change in distinctiveness (consistently positive or consistently negative) holds across different t-thresholds and that the significance of effects is largely maintained across t-thresholds. However, as in lateral VTC, the t-threshold affects the result for some categories: The analysis reveals a significant effect of changes in distinctiveness for faces in the left hemisphere for t values selected by t-threshold >1 and $t>2$. This effect is in line with the analysis using all voxels in medial VTC, which also shows this effect. Likewise, the change in distinctiveness for houses in

the left hemisphere is also significant for t-thresholds >1 and >2 and again this effect mirrors the significant effect for houses in the left hemisphere when using all voxels in medial VTC. Together these analyses suggest that the increase in distinctiveness for faces and houses in left medial VTC may be driven by voxels with rather low (but positive) t-values. Two more changes are found when selecting voxels with t-threshold > 5: Here the increase in distinctiveness for houses in the right hemisphere is no longer significant and there is a significant decrease in distinctiveness for cars.

We have added a short description in the text linking to these new analyses: *“Result were largely the same for a range of thresholds to define selective voxels (Fig. S4).”*

Reviewer #2 (Remarks to the Author):

The present manuscript reports a functional MRI study addressing the developmental changes in the categorical object representation in the visual ventral temporal cortex (VTC), between 5 and 13 years of age. Focusing on the lateral aspect of the VTC, the study measures the distinctiveness between categories: how much the neural response pattern for a category differs from the neural response patterns for other categories. Results show that 1) distinctiveness changes (increases or decreases) over time for some categories but not for others, 2) changes are driven by voxels in category-selective clusters of the lateral VTC, and 3) distinctiveness for faces and words in the lateral VTC correlates with, and predicts, behavioral performance in face recognition and pseudoword respectively.

- Does the work support the conclusions and claims, or is additional evidence needed?
- Are there any flaws in the data analysis, interpretation and conclusions? - Do these prohibit publication or require revision?
- Is the methodology sound? Does the work meet the expected standards in your field?
- Is there enough detail provided in the methods for the work to be reproduced

The manuscript is well written; it tackles an important question with sound methodology and analyses. The results are statistically reliable and interesting. Methods and analyses are described with enough detail for replication.

We thank the reviewer for their positive evaluation of our research.

However, there are some important issues that need to be considered before the implications of the results and the general impact of the work for understanding development can be evaluated. First, a big part of the data (the medial VTC) and of the categories (inanimate objects) are not considered, and it is not clear why. The analysis in the medial VTC could clarify the effects for categories that, in the current analyses, yield inconclusive null results. Second, the results support the strong claim that developmental changes in visual categorical object representation are driven by changes in selective voxels, as opposed to more distributed cortical representations. This seems contrary to what has been suggested by other results in infants (Spriet et al., 2022; 119.8: e2105866119). In addressing a possible conflict, the authors may need to recognize that the developmental change described here may be less general than it may appear, possibly limited to a certain temporal interval (or developmental stage: late childhood to adolescence). Third, there are a number of effects that are not so straightforward and are not explained (e.g., effect of age in the relationship between face distinctiveness and recognition; lateralization of changes in the number representation), and methodological choices that need to be better justified (e.g., definition of selective voxels, variables in the correlation). Below, I discuss these and a few other issues, hoping my comments will help the authors improve the manuscript.

1. First, it is not clear why the study focuses on the lateral VTC, leaving aside the medial VTC. The lateral VTC hosts selective regions for face, limbs and words, but selectivity for other categories (notably, objects and places) can rather be found in the medial VTC (see e.g., Grill-Spector & Weiner, 2014). For categories such as cars, corridors and instruments, the study reports null effects (Fig. 1-2), which are inconclusive –and for no reason, given that data are available. Do those null effects reflect genuine category-specific differences in terms of developmental changes (neural representations for certain categories, but not others, continues to evolve in late childhood/adolescence)? Or do they simply reflect poor measurement in regions that might show effects for those categories of inanimate objects? Developmental changes of face and word representation, which is the main goal of the study, can be better understood by comparison with other categories.

We agree with the reviewer that examining the development of distinctiveness in medial VTC contributes to our understanding of the results presented in the manuscript. Therefore, we have added analyses showing development of distinctiveness in medial VTC to the manuscript. These new figures can be found in new Figures 1e and 2b, as well as in the supplemental Figures S7 & 8. Please see our detailed response to the comment 2 of reviewer 1, who raised a similar concern. Our response also includes the new figures.

In brief, the analysis of distinctiveness in medial VTC reveals a bilateral increase in distinctiveness for houses, as well as an increase in distinctiveness for faces in the left hemisphere. As such, these analyses show additional evidence for differential development of category representation in VTC. That is, we find development of distinctiveness for some categories (faces, words, numbers, limbs and houses), but there is no evidence for development of distinctiveness of other categories (like cars, corridors, string instruments or bodies) in either lateral or medial VTC.

2. One intriguing result of the study is the lack of evidence for developmental changes (in distinctiveness) in non-selective voxels. Is it possible that the time window in which the effects have been tested (late childhood to adolescence) is not a sensitive period to observe such changes? Recent studies suggest a quite different story. Spriet et al. (2022 PNAS) tested the developmental changes in visual object categorization using eight different categories, in infants from 4 months to 18 months. One of the findings is that, as infants grow older, they show (behaviorally) to represent more and more categories and this change is correlated with categorical object representation in ever larger aspects of the ventral cortex. Thus, this study suggests that the development of object category representation proceeds with an increase in the distributedness of the representation. Of course, children/adolescents and infants are different “species” but that’s the question. In concluding for one hypothesis (e.g., change in category distinctiveness is driven by selective voxels), it should be considered that the current time window might be sensitive to only some effects, while other effects (e.g., changes in more distributed representations) may occur at a different time. What kind of development is the development measured in late childhood/adolescence? How does it relate to developmental phenomena occurring earlier (in early infancy)? This speaks to the generality of the current results with respect to explaining the mechanisms of longitudinal development of visual category representation.

We thank the reviewer for raising this point. We believe that our data are not in conflict with the results of Spriet et al. (2022). First, while our analyses presented in Fig. 2 reveal that the changes in distinctiveness appear to be driven mainly by the union of the selective voxels, our results show these age-related changes in distinctiveness when examining all voxels in lateral and medial VTC, which is in

line with the results of Spriet et al. (2022). Second, our study does not make claims about development outside of the tested age range. For instance, in the abstract we write: *“These results suggest that the development of distributed VTC representations has behavioral ramifications and advance our understanding of prolonged cortical development during childhood”*. As such, we limit our claims to the age range of childhood and adolescence.

However, we agree with the reviewer that it is possible that different mechanisms may be at place in infancy and that it is relevant to examine these effects in earlier age ranges. Accordingly, we now mention these considerations in the discussion:

“These results also highlight the importance of examining the development of category representations in other developmental phases, in particular during infancy³⁹⁻⁴² and testing how these developments relate to other behaviors, such as eye gaze patterns toward visual categories⁴³.”

3. To test the link between cortical development and behavioral development concerning face and word processing, the authors carried out correlational analysis considering performance in face recognition/word reading and the distinctiveness of those categories. However, if the goal is to test the link between the two developments (in cortical representation and in behavioral performance), why not consider the changes in category distinctiveness (to be clear, the measure plotted for example in Fig. 1d), rather than the distinctiveness itself. This analysis would support the authors’ conclusion that *“developmental improvements in reading and face recognition abilities are linked to increases in word and face distinctiveness”* (lines 426-427).

We performed the analysis proposed by the reviewer which is interesting. Results shown in the figure below indicate that there is a significant correlation between age-related changes in reading performance and age-related changes in pseudoword distinctiveness over the union of the selective voxels in left lateral VTC (Figure below-left), but there is no significant correlation between age-related changes in face recognition CMFT performance and age-related changes in adult-faces distinctiveness over the union of the selective voxels in right lateral VTC (Figure below-right). The results for words are consistent with the data in Fig 4. The reason that this relationship is not significant for faces is because we have a narrow dynamic range of performance changes in face recognition abilities across our child participants which makes the data difficult to interpret (Figure below-right). Thus, future longitudinal research including younger children and perhaps over wider age spans (e.g., into young adulthood) would allow making behavioral measurements with a larger dynamic range to test these relationships more robustly for faces.

Relationship between change in performance over age vs. change in distinctiveness with age. *Left:* Age-related changes in reading pseudowords vs. age-related changes in pseudoword distinctiveness over the union of selective voxels in left lateral VTC. *Right:* Age-related changes in face recognition (CFMT) performance vs. age-related changes in distinctiveness for faces over union of selective voxels in right lateral VTC.

4. The relationship between word distinctiveness and reading was independent from the effect of age, but the relationship between face distinctiveness and face recognition was dependent on age (the effect was no longer significant in the left hemisphere and there was only a trend in the right hemisphere). On this analysis, the authors conclude that distinctiveness of face representations in lateral VTC predicts face recognition performance. How does this conclusion account for the fact that the effect is not significant when age is considered?

We thank the reviewer for pointing this out. We agree that the result regarding the link between face recognition and distinctiveness for faces is limited as it is only marginally significant when age is added to the model and does not survive correcting for multiple comparisons. We have now highlighted this limitation in the discussion as follows:

“These longitudinal measurements are consistent with our prior cross-sectional data that found that better reading ability in adults than children is coupled with higher distinctiveness in word-selective voxels¹⁸ but also show that the link between reading and distinctiveness for words in the union selective voxels of left lateral VTC was higher than for face recognition and face distinctiveness in the union of selective voxels of right VTC. We hypothesize that the more restricted range of face recognition than reading performance combined with smaller individual variability in the rate of face recognition performance changes over time may contribute to these differences. Future longitudinal studies that include participants with both a larger range of ages and larger range of face recognition abilities (comparable to the range of reading abilities) will be important to tease apart these possibilities.”

5. For the second analysis, the authors separate selective and non-selective voxels. First, why did the authors select as a criterion, a voxelwise effect of $t > 3$? Is this a common/standard criterion for voxelwise selectivity? And how did the authors treat those voxels that were selective ($t > 3$) for more than

a category? And which of the 10 categories had selective voxels in the lateral VTC? How was the distribution of those voxels? Do they give rise to the classic category-selective regions that one would find with a functional localizer? A good sanity check would be to show category-selective clusters on the brain...

We agree with the reviewer that the selection of the t-value as a criterion for defining category-selective voxels this is an important point. We address each of the raised questions below.

- Why did the authors select as a criterion, a voxelwise effect of $t > 3$? Is this a common/standard criterion for voxelwise selectivity?

A threshold of a t-value > 3 was used due to the following reasons. First, our prior research has shown that category-selective regions can be defined reliably in individual subjects including both children and adults using this threshold (Finzi et al., 2021; Gomez et al., 2017, 2018; Natu et al., 2016; Nordt et al., 2021). In addition, a t-value of 3 shows that the effect is 3 standard deviations from the mean. We have added a justification of the threshold to the methods section of the manuscript.

In addition, we have added an extensive control analysis in which we repeat the analyses presented in the second analyses but for voxel sets defined by t-values varying from $t > 1$ to $t > 5$ (new Supplemental Fig. S3 & S4). These results suggest that the findings presented in Fig. 2 are largely robust towards the tested variations in the t-threshold: For most categories we see (i) the same sign of change in distinctiveness (consistently positive or consistently negative) across different t-thresholds and (ii) that significance of effects is largely maintained across t-thresholds even if the numeric values of the change in distinctiveness are not identical. There are also some instances, where the t-threshold affects the change in distinctiveness, which we list in detail in the response to reviewer 1's comment 9.

- And how did the authors treat those voxels that were selective ($t > 3$) for more than a category?

To define the union of category-selective voxels we first defined the selective voxels for each category and then took the union of all of these voxels. This will include voxels that are selective ($t > 3$) for more than one category.

- And which of the 10 categories had selective voxels in the lateral VTC?

We have examined the number and development of selective voxels in lateral VTC in our prior publications. The figure below shows the nr of selective voxels ($t > 3$) in left and right lateral VTC for a group of 5-9-year-old children (lighter colors) and a group of 13-17-year-old teenagers (darker colors) for the 10 categories (Nordt et al., 2021). This figure reveals that for some categories, like words, limbs, bodies or faces, there is a large number of selective voxels, while there are a few selective voxels for other categories, like cars, corridors, or string instruments.

Editorial Note: The figure on this page reproduced with permission from Nordt, M., Gomez, J., Natu, V.S. *et al.* Cortical recycling in high-level visual cortex during childhood development. *Nat Hum Behav* 5, 1686-1697 (2021).

From Supplemental Figure S3, Nordt et al 2021. Acronyms: N = numbers; W = words; L = limbs; B = bodies; AF = adult faces; CF = child faces; C = cars; SI = string instruments; H = houses; Cor = corridors

- How was the distribution of those voxels? Do they give rise to the classic category-selective regions that one would find with a functional localizer?

The union of the selective voxels include both the classic category selective regions, so clustered voxels that show selectivity for a given category, as well as scattered category-selective voxels that pass the threshold ($t > 3$) that are not part of a cluster. Most of the category-selective voxels for faces, words, limbs, and houses are clustered in lateral VTC in both children and adults, consistent with other studies [e.g., Golarai 2007; 2010; 2015; Scherf 2007; Peelen 2009; Meissner 2019; Cohen 2019).

6. From Fig. 1 and 2, changes for numbers are stronger in the right hemisphere. However, the MDS embeddings with unions of selective voxels (Fig. 3) show that representations of pseudowords and numbers increases from age 5 to age 17, particularly in the left hemisphere. Isn't there a conflict?

We thank the reviewer for raising this point and we agree that the reporting of results in Fig 3 can benefit from a more detailed description. While the bar plots in Fig. 1 and 2 represent the change in distinctiveness across the whole sample, the MDS embeddings show data of two subgroups of our sample. Precisely, the MDS embeddings show data of a group of 5-9-year-old children (n=16) and of a group of 13-17-year-old teens (n=13). These depictions of the MDS embeddings were chosen to illustrate the changes in the MDS embeddings with age, they are not used for any statistical analyses. All statistics are run using LMM and using data of the whole sample (full stats are reported in Table 3 & 4). As such, it is possible that the change depicted in the MDS embedding may deviate slightly from the change indicated by the slopes in Fig. 2. To further clarify this point in the manuscript, we have added more information to the legend of Fig. 3. The respective part now reads:

"Multidimensional scaling (MDS) embeddings for the category representation in different subsets of voxels for two age groups: 5-9-year-olds (n=16 participants, small circles) and 13-17-year-olds (n=13 participants, larger circles). These age groups are used to illustrate the change in the representational space and are based on average RSMs of children in two age groups. All statistics are run using the full sample (Fig. 2, Fig. 3CF)."

7. Limbs: distinctiveness decreases over time when considering all the voxels (Fig. 1) but then this effect is significant neither in selective or in non-selective voxels.

What does this mean? Could it be the case that distinctiveness for this category is contributed by both selective and non-selective voxels?

The reviewer raises an interesting point. We agree that it is possible that the decrease in distinctiveness for limbs comes from both the union of the selective and the non-selective voxels. In fact, our new analyses presented in Fig S3, where we repeat analyses shown in Fig 2 but vary the t-threshold to define selective voxels, points to a similar result:

This plot replicated here on the right, illustrates a significant decrease in distinctiveness for limbs in the right hemisphere over all voxels in lateral VTC that showed selectivity to limbs at $t > 1$. In contrast to the threshold of $t > 3$ (used in the main analysis), this is a low threshold and includes weakly selective voxels. This suggests that voxels with weak selectivity to limbs contribute to the observed decrease in distinctiveness.

We have added a discussion of this finding in the manuscript:

“Interestingly, the decrease in distinctiveness for limbs that was observed across all voxels in lateral VTC, was not significant when examining the union of the selective voxels or the non-selective voxels as separate subsets. This suggests that both subsets of voxels may contribute to the decrease in distinctiveness for limbs.”

8. From Fig. 2 it looks like distinctiveness for child faces increases more in left than in the right hemisphere. Is this statistically true? If so, does this result matter?

It is true that the increase in distinctiveness for both face types is numerically stronger in the left compared to the right hemisphere in the union of the selective voxels, which seemingly contradicts with the typically observed right-hemisphere advantage observed for face processing. However, when examining the underlying data (see plots below), and not just the slopes, these show that distinctiveness for faces is both (i) overall higher in the right compared to the left hemisphere (see means and SD in the plots) and (ii) that distinctiveness for faces already starts higher in 5-year-olds in the right hemisphere compared to the left. As such, these data confirm the lateralization of faces to the right hemisphere and suggest that the development of face distinctiveness in the right hemisphere may have started earlier than that in the left hemisphere.

Please see also our response to reviewer 3's question 4, where we include the new paragraph in the manuscript discussing observed lateralization effects in the present dataset.

9. Missing words in line 402?

Thank you for pointing this out. We have added the missing word and the full sentence now reads: *"That is, better face recognition performance was associated with higher values of distinctiveness for faces (Fig. 4C)."*

Reviewer #3 (Remarks to the Author):

This is a fascinating dataset and an equally intriguing set of hypotheses. The links made between changes in distinctiveness and recognition behaviour is a welcome one and bolsters the claims made. I am enthusiastic about the data, but have a number of queries that I feel would only strengthen an already strong manuscript, which I outline below (in no particular order).

We thank the reviewer for their positive summary of our research.

1. Can the authors confirm that analyses were performed on voxels and not surface nodes (or indices if you prefer)? Fig 1.b shows a surface representation of left lateral VTC. If this is for illustrative purposes only, can it be stated thus?

Yes, the central analyses were performed on voxels, not on surface nodes. The anatomical ROIs (lateral and medial VTC) were defined on the surface representation of each participant as this representation allows for a better visualization of the anatomical landmarks based on which we marked the ROIs in individual participants. Therefore, to show these ROIs and the underlying anatomical landmarks we chose to depict an example ROI on the cortical surface representation in Fig. 1b. To clarify this issue in the manuscript, we have added the sentence “The following analyses were performed in voxel space (not on surface nodes).” In the methods section under *Analysis of functional MRI data*.

2. Unless I missed it, were off diagonal values averaged together in the calculation of distinctiveness (Fig 1.a)?

Yes, we computed the average of the off-diagonal values in the calculation of distinctiveness. (on-diagonal minus the mean of all off-diagonal values (excluding the other category from same domain)).

3. Indicators of significance would be useful in Fig. 1d

We have added asterisks indicating significant effects (after FDR correction).

4. The lack of hemispheric difference in the distinctiveness of both child and adult faces was a surprise to me. The left hemisphere preference for pseudowords is clearly present, but I would have expected a commensurate change in the right hemisphere. The authors do not discuss this and its implications for the debate about lateralisation of function and I believe this is a missed opportunity.

We agree with the reviewer that this is an interesting point. We have included the following discussion of the aspect of lateralization in the discussion section:

“The present data also offers insight on the laterality of the development of category distinctiveness. In particular, prior research has suggested a graded developmental lateralization of word representations to the left hemisphere^{11,18,31,34} and of face representations to the right hemisphere^{7,31,34}. Consistent with this hypothesis, we see that the development of distinctiveness for pseudowords was significant only in the left hemisphere and that distinctiveness in the left, but not right hemisphere, predicts reading performance. With regard to the lateralization of face representations, our results reveal a bilateral development of distinctiveness across all voxels in lateral VTC, as well as across the union of category selective voxels, in line with prior studies finding that the number of face-selective voxels increases across development in both hemispheres²³. It should be highlighted at this point that the results shown in Figs. 1 and 2 summarize the change in distinctiveness with age, rather than the level of face distinctiveness. That is, while the change in face distinctiveness was bilateral, distinctiveness for faces was nonetheless numerically higher in the right than left hemisphere. As such, these results support lateralization of word and face representations in children and teens, but suggest that development is not restricted to the dominant hemisphere.”

5. Paragraph starting lines 178: The authors mention that motion was added as a predictor into the

LMM, but weren't the values that go into the LMM already computed on data that had been motion corrected (see methods)? This needs to be clarified.

Yes, the motion predictor was added into the LMM even though we applied both within and between-run motion to the data. The reason for testing if motion is a significant predictor in addition to performing motion correction is that motion correction is imperfect, and we wanted to ensure that developmental effects are not driven by motion induced quality differences that may still be present in the data after performing motion correction.

We have added a clarifying sentence to the manuscript:

"To ensure that developmental effects are not driven by differences in scan quality across age, we first tested whether motion during scanning and timeseries signal-to-noise ratio (tSNR) contribute to measures of distinctiveness. Checking for effects of motion was included after our data had already been corrected for motion (see Methods) to further ensure that our results are not impacted by motion-related artifacts that are missed by motion correction algorithms."

6. The analysis of selective V non-selective requires deeper investigation and broader discussion:

a. Why was a t-value of 3 selected?

A threshold of a t-value > 3 was used due to the following reasons. First, our prior research has shown that category-selective regions can be defined reliably in individual subjects including both children and adults using this threshold (Finzi et al., 2021; Gomez et al., 2017, 2018; Natu et al., 2016; Nordt et al., 2021). In addition, a t-value of 3 shows that the effect is 3 standard deviations from the mean. We have added a justification of the threshold to the Methods section of the manuscript.

b. What p-level does this reflect?

A t-value of 3 reflects a p-Value of 0.0014 (two-tailed testing)

c. If changes in distinctiveness are truly only present within category-selective voxels, does the degree of distinctiveness increase as selectivity increased (e.g., if a voxel is selective at $t=6$ is category representation more distinctive than a voxel at $t=3$). Again, this feels like a missed opportunity to fully explore these effects and provide a comprehensive picture.

d. At what levels of selectivity does distinctiveness break down? Would a voxel at $t=2.99$ really not show distinctiveness?

The reviewer raises an important point. To address this issue, and similar concerns raised by the other reviewers, see our detailed response to this point in the response to reviewer 1's comment 9.

In brief, we have conducted an analysis in which we examine the development of distinctiveness for the union of selective voxels defined by varying t-value thresholds from $t>1$ to $t>5$. We have added the results of this analysis as new Fig S3 & S4. In addition, the figures showing these results can be found in our response to reviewer 1's comment 9. These results suggest that the findings presented in Fig. 2 are largely robust towards the tested variations in the t-threshold: For most categories we see (i) the same sign of change in distinctiveness (consistently positive or consistently negative) across different t-thresholds and (ii) that significance of effects is largely maintained across t-thresholds even if the numeric values of the change in distinctiveness are not identical. There are also some instances, where the t-threshold affects the change in distinctiveness, which we list in detail in our response to reviewer 1's comment 9.

7. Related to point 4 above, the analysis of distinctiveness in category-selective voxels (Fig. 2) shows

higher levels of average distinctiveness in the left hemisphere over the right for both adult and child faces – this needs to be at least highlighted and discussed.

We thank the reviewer for pointing this out. While our data in Fig 2a show that there is a numerically stronger increase for distinctiveness for faces in the left compared to the right hemisphere in the union of the selective voxels, levels of distinctiveness for faces are overall higher in the right compared to the left hemisphere (for the plots showing these data, see our response to reviewer 2's comment 8). We agree with the reviewer that this finding may be surprising at first sight and warrants a discussion. Therefore, in the manuscript we now discuss these effects as follows:

“With regard to the lateralization of face representations, our results reveal a bilateral development of distinctiveness across all voxels in lateral VTC, as well as across the union of category selective voxels, in line with prior studies finding that the number of face-selective voxels increases across development in both hemispheres²³. It should be highlighted at this point that the results shown in Figs. 1 and 2 summarize the change in distinctiveness with age, rather than the level of face distinctiveness. That is, while the change in face distinctiveness was bilateral, distinctiveness for faces was nonetheless numerically higher in the right than left hemisphere.”

8. Again, I remain enthusiastic about the manuscript, but too much is made of the difference between the prediction error of selective V non-selective voxels ($p=0.045$).

We agree that this difference should be presented and interpreted with caution. Therefore, we now highlight that the effect is small in the results section:

“Indeed, we found a small but significant effect when comparing the prediction error for the model using the distinctiveness over the selective voxels vs that over the non-selective voxels (Fig. 4B, swarm plot, two-sided t-test comparing the difference in error to zero: $t(25)=-2.11$, $p=0.045$).”

9. Paragraph beginning line 397:

a. There is an incomplete sentence on line 400-401 that needs rectifying.

Thank you for pointing this out. We have added the missing word and the full sentence now reads: *“That is, better face recognition performance was associated with higher values of distinctiveness for faces (Fig. 4C).”*

b. There needs to be a discussion of the fact that when age is added for to the LMM this result is no longer significant at $p < .05$. This is important because the authors ultimately put forward the case for distinctiveness being restricted to selective voxels, but that finding appears only to be consistent for pseudowords.

Thank you for pointing this out. We have now highlighted the limitation of the link between distinctiveness for faces and face recognition in the discussion as follows:

“In particular, the link between face recognition performance and distinctiveness for faces over the union of selective voxels in the right hemisphere was no longer significant when age was added to the model. We hypothesize that the more restricted range of face recognition than reading performance combined with smaller individual variability in the rate of face recognition performance changes over time may contribute to these differences. Future longitudinal studies that include participants with both a larger

range of ages and larger range of face recognition abilities (comparable to the range of reading abilities) will be important to tease apart these possibilities.”

c. There are no corrections for multiple comparisons applied to the results reported.

Thanks for this suggestion. We have now added FDR-correction for the tests performed on the links between behavior and distinctiveness for each category.

The link between reading performance of pseudowords and distinctiveness for pseudowords remains significant after FDR-correction:

“We found a significant and positive relationship between reading performance of pseudowords and distinctiveness for pseudowords over the union of the selective voxels of left lateral VTC (Fig. 4A, $\beta_{distinctiveness}=40.19$, $t(62)=2.94$, $p_{FDR}=0.016$, LMM, random slope and intercept across participants).”

Likewise, the link between distinctiveness for pseudowords and reading in the model that also includes age survives FDR-correction:

“The effect of distinctiveness predicting reading performance remained significant when age was added to the LMM ($\beta_{distinctiveness}=28.92$, $t(61)=2.4$, $p_{FDR}=0.046$; $\beta_{age}=2.45$, $t(61)=3.44$, $p_{FDR}=0.007$), showing that the effect of distinctiveness was independent from the effect of age.”

The link between face recognition (performance on CFMT adults) and distinctiveness for adult faces remains significant after FDR correction:

“Likewise, distinctiveness for adult faces in the union of selective voxels of right lateral VTC was significantly and positively related to face recognition performance ($\beta_{distinctiveness}=31.49$, $t(80)=3.82$, $p_{FDR}=0.0008$, LMM with random slope and random intercept across participants).”

However, once age is added to the model, the effect is no longer significant, only trending:

“When age was added to the LMM the effect of distinctiveness was only trending and was no longer significant after correction for multiple comparison ($\beta_{distinctiveness}=16.41$, $t(79)=1.97$, $p_{FDR}=0.08$; $\beta_{age}=3.94$, $t(79)=7.89$, $p_{FDR}<0.001$).”

Nonetheless, distinctiveness for faces still is important for behavior as an independent analysis using a leave-one-out-subject approach reveals that face recognition performance in the left out subject can be predicted from their faces distinctiveness with an error $\sim 11\%$, which is pretty striking. We write:

“Notably, distinctiveness for adult faces over the selective voxels of right VTC predicts face recognition performance in left out subjects (LOOCV, Fig. 4D-purple boxplot). Face recognition ability is predicted from face distinctiveness over the selective voxels of right lateral VTC in the left-out subject with a median prediction error of 10.93%. Adding age to the model did not significantly reduce the error ($t(28)=-0.42$, $p=0.68$, Fig. 4D-dark gray boxplot) and for most subjects, the prediction error was higher from a model based on the non-selective voxels than a model based on the union of selective voxels (Fig. 4D-light gray boxplot and swarm plot, t-test comparing the difference in error to zero: $t(28)=-3.19$, $p=0.004$).”

REVIEWER COMMENTS

Reviewer #1 (Remarks to the Author):

I appreciate the thorough and well considered responses to my own and the other reviewers' comments. I am satisfied with the answers provided, and only have a few minor final comments/questions.

> Page 19: Together these data provide evidence that developmental changes in the degree of selectivity affect distributed responses. That is, developmental increases in category selectivity lead to increases in category distinctiveness and developmental decreases in selectivity lead to decreases in distinctiveness.

I would consider this a plausible possibility based on these combined data, but am missing direct evidence. For this I would expect to see that people with a larger change in category selective voxels also show a larger change in category distinctiveness, but I believe only evidence that both processes occur with age is available - so I would say the data merely suggest or are consistent with such a relationship.

> I found the results of the new "disk" analysis interesting. As the authors note, it adds to the previous analyses by showing that changes in category distinctiveness within the unity of category selective voxels are driven by voxels selective for the corresponding category but also by voxels selective for other categories. I missed a consideration of the implications of this finding, for example when referring to the debate about modular vs. distributed category representations (line 644).

> It is not clear to me why the lateral and new medial VTC were kept separate in the analysis. Merging them seems simpler and appropriate for testing distributed representations?

> I also noted one detail I hadn't noticed before that would benefit from some clarification:

- distinctiveness ranges from 0-2 (page 5) - please clarify why some values in Fig 1C nevertheless go below zero. I think it would be helpful to add a supplement with formulas used to compute distinctiveness for each category as different approaches were used across categories, and with a clarification of the expected range of values and possible deviations.

Reviewer #2 (Remarks to the Author):

I confirm my previous positive evaluation of the work of Nordt et al. The new version of the manuscript provides a new set of analyses and integrates new points of discussion, which strengthen the importance of this work. This work has the merit of contributing to the understanding of the developmental aspects of visual object categorization, providing both theoretical advances with respect to the debates on the distributedness of category-information in the visual cortex and related developmental changes, and methodological advances illustrating a valid way to study these issues. Therefore I would like to recommend the publication of the present version of the manuscript, with no further reservations.

Reviewer #3 (Remarks to the Author):

The authors have addressed my concerns and I remain enthusiastic about the data and manuscript.

Point-by-point response

We thank the reviewers for their positive feedback on our previous work and their constructive questions. Our responses to the reviewer's comments are indicated in blue under each comment. We have highlighted changes in the manuscript using the track changes function.

REVIEWER COMMENTS

Reviewer #1 (Remarks to the Author):

I appreciate the thorough and well considered responses to my own and the other reviewers' comments. I am satisfied with the answers provided, and only have a few minor final comments/questions.

> Page 19: Together these data provide evidence that developmental changes in the degree of selectivity affect distributed responses. That is, developmental increases in category selectivity lead to increases in category distinctiveness and developmental decreases in selectivity lead to decreases in distinctiveness.

I would consider this a plausible possibility based on these combined data, but am missing direct evidence. For this I would expect to see that people with a larger change in category selective voxels also show a larger change in category distinctiveness, but I believe only evidence that both processes occur with age is available - so I would say the data merely suggest or are consistent with such a relationship.

We thank the reviewer for this suggestion and we agree that a more careful phrasing is appropriate. Therefore, we have adjusted these sentences, which now read:

“Together these data are consistent with the idea that developmental changes in the degree of selectivity affect distributed responses. That is, these data suggest that developmental increases in category selectivity lead to increases in category distinctiveness and developmental decreases in selectivity lead to decreases in distinctiveness.”

> I found the results of the new "disk" analysis interesting. As the authors note, it adds to the previous analyses by showing that changes in category distinctiveness within the unity of category selective voxels are driven by voxels selective for the corresponding category but also by voxels selective for other categories. I missed a consideration of the implications of this finding, for example when referring to the debate about modular vs. distributed category representations (line 644).

We agree with the reviewer that it is an interesting aspect to mention in the discussion. The respective paragraph now reads:

“The present results also have important theoretical implications for the debate on how distributed¹³ vs. modular²⁸ category representations in VTC contribute to behavior. On the one hand, our data reveal that representations of animate (faces and bodies) categories are separable from representations of the other inanimate categories in both selective and non-selective voxels (Fig. 3), consistent with the predictions of the distributed hypothesis¹³. On the other hand, we find significant development of category distinctiveness across the union of the selective voxels in lateral VTC, but no evidence for development of category distinctiveness in the non-selective voxels, contrary to the predictions of the distributed hypothesis¹³. Similarly, the analysis of the development of distinctiveness in disk ROIs that are selective to one particular category (Fig. S5) showed that distinctiveness developed for the category used to define the ROI (i.e., distinctiveness for words developed in the word-selective ROI) but also that development was not limited to the ROI-defining category.”

> It is not clear to me why the lateral and new medial VTC were kept separate in the analysis. Merging them seems simpler and appropriate for testing distributed representations?

We thank the reviewer for raising this point. While we understand the reasoning for combining lateral and medial VTC to investigate category distinctiveness, we think it is important to separate VTC into its lateral and medial partitions due to the following reasons:

First, there is accumulating evidence that medial and lateral VTC develop differently as the development of lateral VTC is particularly prolonged (Golarai et al., 2007; Scherf 2007; Gomez et al., 2017; Nordt et al., 2021). Second, medial VTC shows a peripheral bias but lateral VTC shows a foveal bias (Levy et al., 2001; Weiner et al., 2014) and there is evidence that foveal regions continue to develop during this age range (e.g., Gomez et al., 2018).

Thus, combining lateral and medial VTC would not allow us to test if there are also developmental differences in distributed responses in VTC. For instance, our results show that distinctiveness for words develops in left and distinctiveness for numbers develops in right lateral VTC, but we find no evidence for a development in medial VTC in line with the above-mentioned findings on eccentricity bias and differential development of VTC partitions. Without the separation into lateral and medial VTC we would have lost the ability to detect the differential development of distributed representations across lateral and medial VTC.

> I also noted one detail I hadn't noticed before that would benefit from some clarification:

- distinctiveness ranges from 0-2 (page 5) - please clarify why some values in Fig 1C nevertheless go below zero. I think it would be helpful to add a supplement with formulas used to compute distinctiveness for each category as different approaches

were used across categories, and with a clarification of the expected range of values and possible deviations.

We thank the reviewer for highlighting this aspect because this was a typo in the prior version of the manuscript. The range of distinctiveness is from -2 to 2 and as the reviewer notices, it includes negative values. For instance, if the within-category correlation was -1 and the between-category correlation was 1, the resulting distinctiveness would be -2, the lowest distinctiveness value. We have corrected this number in the text.

The same approach to compute distinctiveness was used for all categories. A detailed description of how category distinctiveness is computed can be found in the methods:

“The distinctiveness of a category is defined as the within-category minus between-category similarity of distributed responses leaving out the between-category similarity with the other category from the same domain. This is illustrated in the gray box in Fig. 1A: In the RSM the values on the diagonal represent the within-category similarity, while the off-diagonal values represent the between-category similarities. For instance, to compute the distinctiveness for words, the between-category similarities for all categories except numbers are subtracted from the within category similarity for words. The reason for leaving out the other category from the same domain is that for some domains the two categories are very similar to each other (such as adult and child faces) while for other domains the two categories differ from each other to a greater extent (such as cars and string instruments) so we chose a procedure that would be least biased and will not differently affect stimuli from various domains. “

Reviewer #2 (Remarks to the Author):

I confirm my previous positive evaluation of the work of Nordt et al. The new version of the manuscript provides a new set of analyses and integrates new points of discussion, which strengthen the importance of this work. This work has the merit of contributing to the understanding of the developmental aspects of visual object categorization, providing both theoretical advances with respect to the debates on the distributedness of category-information in the visual cortex and related developmental changes, and methodological advances illustrating a valid way to study these issues. Therefore I would like to recommend the publication of the present version of the manuscript, with no further reservations.

Reviewer #3 (Remarks to the Author):

The authors have addressed my concerns and I remain enthusiastic about the data and manuscript.